# Statistical Guarantees of Distributed Nearest Neighbor Classification

**Jiexin Duan**
Department of Statistics
Purdue University
West Lafayette, Indiana, USA
duan32@purdue.edu

**Xingye Qiao**
Department of Mathematical Sciences
Binghamton University
New York, USA
qiao@math.binghamton.edu

**Guang Cheng**
Department of Statistics
Purdue University
West Lafayette, Indiana, USA
chengg@purdue.edu

## Abstract

Nearest neighbor is a popular nonparametric method for classification and regression with many appealing properties. In the big data era, the sheer volume and spatial/temporal disparity of big data may prohibit centrally processing and storing the data. This has imposed considerable hurdle for nearest neighbor predictions since the entire training data must be memorized. One effective way to overcome this issue is the distributed learning framework. Through majority voting, the distributed nearest neighbor classifier achieves the same rate of convergence as its oracle version in terms of the regret, up to a multiplicative constant that depends solely on the data dimension. The multiplicative difference can be eliminated by replacing majority voting with the weighted voting scheme. In addition, we provide sharp theoretical upper bounds of the number of subsamples in order for the distributed nearest neighbor classifier to reach the optimal convergence rate. It is interesting to note that the weighted voting scheme allows a larger number of subsamples than the majority voting one. Our findings are supported by numerical studies.

## 1 Introduction

Classification is one of the pillars of statistical learning. The nearest neighbor classifier is among the conceptually simplest and most popular of all classification methods. However, it is a memory-intensive method in that the entire training data must be memorized to make a prediction. Instead of spending long time to learn a simple rule from the training data, the nearest neighbor classifier defers the computational burden to the prediction stage. The asymptotic properties of the nearest neighbor classification have been studied in [24, 14, 17, 46, 34, 11, 27, 25, 22, 52], among others. [18, 6, 12] provided extensive surveys of the existing literature regarding $k$-NN classifiers.

In the era of big data, due to the unprecedented growth of the sample size and dimension of the data, denoted as $N$ and $d$, the time and space complexities of nearest neighbor methods are huge. A naive algorithm for $k$-nearest neighbor ($k$NN) classification would compute distances from each query point to all the $N$ training data points, sort the distances, and identify the $k$ smallest distances. Using this naive approach, a single search query has running time between $O(N)$ to $O(N\log(N))$ depending on the efficiency of the sort method [31]. With a large number of training data, having

each search query take $O(N)$ time can be prohibitively expensive. The space complexity for storing the training data is $Nd$. For very large data, $k$NN cannot even be conducted on a single machine if the sample size exceeds the memory of the machine.

A few proposals [1, 41] suggested to distribute the data into multiple local machines and/or leverage a distributed computing environment, for example, Apache Hadoop that uses the MapReduce paradigm, to process high volume data. However, except for organizations that are known for their ability to exploit large data assets, such as high-tech corporations or research institutions, these distributed computing environments are often not very user friendly or accessible to many average users. The irony is that $k$NN is meant to be a simple yet powerful approach that even a layman can comprehend. There are a group of approximate nearest neighbor search algorithms, such as the locality-sensitive hashing methods [32], designed for processing large data sets. However, their implications to the learning performance is less known (with few exceptions such as [26]). Additionally, there are a few techniques that have been empirically used very well, such as the random projection or partition trees [33, 39, 15] and boundary trees [42]. All these trees are used for approximate nearest neighbor search. Some theory [16] has shown that a simplification of the random projection tree may be used for exact nearest neighbor search. Currently they lack more theoretical guarantees. Likewise, [43] proposed scalable nearest neighbor algorithms for high dimensional data, with little statistical guarantee available for the classification performance of the proposed algorithms.

In this article, we study the nearest neighbor classification in a distributed learning framework designed to alleviate the space and time complexity issues in the big data setting. A few recent works for various learning tasks such as regression and principal component analysis have fallen under the distributed learning umbrella, e.g., [56, 13, 3, 57, 23, 36, 47].

We first investigate a type of distributed nearest neighbor classifier (DiNN), in which data are divided to $s$ subsamples (with subsample size $n \ll N$), nearest neighbor predictions are made locally at each subsample, which are aggregated by majority voting (hence, M-DiNN). This simple effort can substantially reduce the time and space complexity[1] and is easy to generalize to other classifiers. [45] has studied M-DiNN and proved the optimal rate of convergence for its regret. However, while it is reassuring to know that the convergence rate is optimal, there is more to say about the learning performance of the M-DiNN method, especially in terms of the multiplicative constant of the regret, in addition to the rate. This has motivated us to pursue a more precise quantification of the regret.

Our foremost contribution is to provide an asymptotic expansion form of the regret of the M-DiNN method. This result is not a trivial extension from that in [46]. Specifically, we need an extra normal approximation by the uniform Berry-Esseen theorem [37] as $s$ diverges, which leads to residual terms that are bounded in a nontrivial way. With carefully chosen weights, the regret of M-DiNN achieves the optimal convergence rate. However, our result reveals that there is some loss in the learning performance, namely, a multiplicative constant loss which depends on the data dimension only, caused by a Taylor expansion of the normal cumulative distribution function at 0; see Remark 1. Such a loss is due to the use of majority voting, and hence is dubbed as the majority voting constant.

[45] studied M-DiNN only, which is now found to be less than optimal due to the majority voting constant loss above. To eliminate this loss, we consider an alternative weighting scheme called weighted voting. [19] introduced weighted voting to construct good ensembles of classifiers, which performed better than the base classifiers. [35] proposed a probabilistic framework for classifier ensemble by weighted voting, and conducted some simulations to show that weighted voting outperformed majority voting under certain conditions. Our second contribution is the proof that DiNN with weighted voting (W-DiNN) achieves exactly the same asymptotic regret as the "oracle" optimal weighted nearest neighbor (OWNN) [46], in terms of both the rate of convergence and the multiplicative constant. (We define an "oracle" classifier as the one trained on a single machine that has infinite storage and computing power and has access to the entire training data.) Moreover, the time and space complexity of W-DiNN are similar to M-DiNN and much smaller than the oracle classifier.

Our third contribution is the identification of *sharp* upper bounds for the number of subsamples in the M-DiNN and W-DiNN classifiers, that we can afford in order for both classifiers to achieve the optimal convergence rate. In practice, these upper bounds can provide some theoretical guidance on how to choose the number of subsamples in DiNN.

Much of our findings in this paper is motivated by the study of optimal weighted nearest neighbor (OWNN) by [46]. The DiNN method seems to resemble the bagged nearest neighbor (BNN) method which was closely studied by [30, 7, 5], except that DiNN uses data divisions and BNN is based on bootstrap (sub)sampling. In addition, the two methods are fundamentally different in terms of their goals. The DiNN method is aimed to deal with big data that cannot be processed by a single machine, while bagging's goal is to improve the stability [8, 55] and classification accuracy.

Previous work [52, 51] tried to speed up the nearest neighbor methods by parallel computing. However, [52] only provide a convergence rate, while we characterize the multiplicative constant. Moreover, their approach was not a true distributed approach. It only moves the computational time from the predicting step to the preprocessing step and reinforces the learning performance by a bootstrap like procedure. [51] is mainly concerned with adversarial learning that includes distributed NN as a special case. Their results also provide convergence rate only.

DiNN can actually work without a parallel computing environment. For example, in a multi-cohort medical study, it is fairly common for multiple institutes to collect sensitive patient data separately. Regulations and privacy issues make it impossible to gather all the patient data at a centralized location. Using a distributed learning idea, to predict the class label for a new instant, one can make a prediction at each institute locally, then combine the results to reach a final prediction, without revealing the information of the training data. The theoretical study we conduct in this work can help to understand the learning performance in this scenario.

The rest of this article is organized as follows. We derive the asymptotic expansion form for the regret of M-DiNN in Section 3, followed by some asymptotic comparisons between M-DiNN and the oracle WNN. In Section 4, we focus on W-DiNN. In Section 5, we conduct some numerical studies to illustrate the theoretical results. Some concluding remarks are given in Section 6.

## 2 Preliminaries

Let $(X, Y) \in \mathcal{R} \times \{0, 1\}$ be a random couple with a joint distribution $P$ where $\mathcal{R} \subset \mathbb{R}^d$. We regard $X$ as a $d$-dimensional vector of features for an object and $Y$ as a label indicating that the object belongs to one of two classes. Denote the prior probability as $\pi_j := \mathbb{P}(Y = j)$ and the conditional distribution of $X$ given $Y = j$ as $P_j$ for $j = 0, 1$. Hence, the marginal distribution of $X$ is $\bar{P} = \pi_1 P_1 + (1 - \pi_1) P_0$. Let $\eta(x) = \mathbb{P}(Y = 1 | X = x)$ denote the regression function. For a classifier $\phi: \mathbb{R}^d \to \{0, 1\}$, its risk is defined as

$$R(\phi) = \mathbb{P}(\phi(X) \neq Y),$$

which is minimized by the Bayes classifier $\phi^*(x) = \mathbb{1}\{\eta(x) \geq 1/2\}$. The corresponding risk $R(\phi^*)$ is thus called the Bayes risk. In practice, a classification procedure $\Psi$ is applied to a training data set $\mathcal{D} := \{X_i, Y_i\}_{i=1}^n$ to produce a classifier $\widehat{\phi}_n = \Psi(\mathcal{D})$, with the corresponding risk $\mathbb{E}_{\mathcal{D}}[R(\widehat{\phi}_n)]$. Here, $\mathbb{E}_{\mathcal{D}}$ denotes the expectation with respect to the distribution of $\mathcal{D}$. The regret of $\Psi$ is defined as:

$$\text{Regret}(\Psi) = \mathbb{E}_{\mathcal{D}}[R(\widehat{\phi}_n)] - R(\phi^*).$$

We consider the general weighted nearest neighbor (WNN) classifier. For a query point $x$, let $(X_{(1)}, Y_{(1)}), (X_{(2)}, Y_{(2)}), \ldots (X_{(n)}, Y_{(n)})$ be the sequence of observations with ascending distance to $x$, and denote $w_{ni}$ as the (non-negative) weight assigned to the $i$-th neighbor of $x$ with $\sum_{i=1}^n w_{ni} = 1$. Define $\widehat{S}_{n, \boldsymbol{w}_n}(x) := \sum_{i=1}^n w_{ni} Y_{(i)}$ as an estimate to the $\eta(x)$. The WNN prediction is

$$\widehat{\phi}_{n, \boldsymbol{w}_n}(x) = \mathbb{1}\left\{\widehat{S}_{n, \boldsymbol{w}_n}(x) \geq 1/2\right\},$$

where $\boldsymbol{w}_n$ denotes the weight vector. When $w_{ni} = k^{-1}$ for $1 \leq i \leq k$, or 0 for $i > k$, WNN reduces to the standard $k$-nearest neighbor ($k$NN) classifier, denoted as $\widehat{\phi}_{n,k}(x)$. As an important starting point of our analysis, Proposition 1 in [46] provides the asymptotic expansion of WNN regret.

**Proposition 1** *(Asymptotic Regret for WNN) Assuming (A1)–(A4) stated in Appendix 1, for each* $\beta \in (0, 1/2)$, *we have, uniformly for* $\boldsymbol{w}_n \in W_{n,\beta}$, *as* $n \to \infty$,

$$\text{Regret}(\widehat{\phi}_{n, \boldsymbol{w}_n}) = \left[B_1 \sum_{i=1}^n w_{ni}^2 + B_2 \left(\sum_{i=1}^n \frac{\alpha_i w_{ni}}{n^{2/d}}\right)^2\right]\{1 + o(1)\}, \tag{1}$$

where $\alpha_i = i^{1+\frac{2}{d}} - (i-1)^{1+\frac{2}{d}}$. *Constants $B_1, B_2$ and $W_{n,\beta}$[2] are defined in Appendix 2.*

We remark that the first term in (1) can be viewed as the variance component of regret, and the second term the squared bias. By minimizing the asymptotic regret (1) over weights, [46] obtained the optimal weighted nearest neighbor (OWNN) classifier.

## 3 DiNN classifier via majority voting

In this section, we analyze the distributed (weighted) nearest neighbor classifiers with majority voting (M-DiNN). Our analysis shows that the regrets of M-DiNN and its oracle counterpart, share the same convergence rate, given that the weights for local classifiers are carefully chosen, and their difference boils down to a multiplicative constant. The main idea of M-DiNN (summarized in Algorithm 1) is straightforward:

- randomly partition a massive data set $\mathcal{D}$ with size $N$ into $s$ subsamples;
- a local WNN classifier is obtained based on each subsample;
- the final classifier is an outcome of the majority voting over the $s$ local WNN predictions.

For simplicity, assume equal subsample size, $n = N/s$, and local WNN classifiers use the same weights $\boldsymbol{w}_n$. We consider the nontrivial setting that $s$ and $n$ grow with $N$: $s = N^\gamma$ and $n = N^{1-\gamma}$.

---

**Algorithm 1** DiNN with majority voting (M-DiNN)

**Input:** Data set $\mathcal{D}$, number of partitions $s$, local weight vector $\boldsymbol{w}_n$ and query point $x$.
**Output:** M-DiNN.
1: Randomly split $\mathcal{D}$ into $s$ subsamples with equal size $n$.
2: **for** $j = 1$ to $s$ **do**
3:     Obtain the WNN classifier $\widehat{\phi}_{n,\boldsymbol{w}_n}^{(j)}(x)$ based on the $j$-th subsample.
4: **end for**
5: Majority voting of all classification outcomes $\widehat{\phi}_{n,\boldsymbol{w}_n}^{(j)}(x)$:

$$\widehat{\phi}_{n,s,\boldsymbol{w}_n}^M(x) = \mathbb{1}\Big\{\frac{1}{s}\sum_{j=1}^s \widehat{\phi}_{n,\boldsymbol{w}_n}^{(j)}(x) \geq 1/2\Big\}. \tag{2}$$

6: **return** $\widehat{\phi}_{n,s,\boldsymbol{w}_n}^M(x)$.

---

Theorem 1 is our first main result. It gives an asymptotic expansion formula for the regret of the M-DiNN classifier given weight vector $\boldsymbol{w}_n$.

**Theorem 1** *(Asymptotic Regret for M-DiNN) Assume the same assumptions as in Proposition 1, and*

$$\sum_{i=1}^n w_{ni}^3 / (\sum_{i=1}^n w_{ni}^2)^{3/2} = o(s^{-1/2}(\log(s))^{-2}). \tag{3}$$

*We have, uniformly for $\boldsymbol{w}_n \in W_{n,\beta}$, as $n, s \to \infty$,*

$$\text{Regret}(\widehat{\phi}_{n,s,\boldsymbol{w}_n}^M) = \Big[B_1 \frac{\pi}{2s}\sum_{i=1}^n w_{ni}^2 + B_2\Big(\sum_{i=1}^n \frac{\alpha_i w_{ni}}{n^{2/d}}\Big)^2\Big]\{1 + o(1)\}. \tag{4}$$

In contrast with Proposition 1, the first term in the asymptotic regret of M-DiNN in Theorem 1 is reduced by a factor of $\pi/(2s)$, while the squared bias term remains the same. This variance reduction effect $1/s$ is not surprising given the study of bagging [8], and has also been observed in the nonparametric regression setup, e.g., [56]. Rather, the appearance of the constant $\pi/2$ is new and motivates another version of distributed classification in Section 4. In this paper, we do not consider the trivial case of fixed $s$ (i.e. $\gamma = 0$), which is almost equivalent to the case of [46]. See [52] for a relevant work.

**Remark 1** *Theorem 1 is not a trivial extension from Proposition 1 as $s$ diverges. Specifically, we need an extra normal approximation of $\mathbb{P}\big(\widehat{\phi}^M_{n,s,\boldsymbol{w}_n}(x) = 0\big)$ by the uniform Berry-Esseen Theorem [37] as $s$ diverges. In fact, the factor $\pi/2$ in (4) comes from a Taylor expansion of the normal cumulative distribution function at 0; see Lemma 1.*

**Remark 2** *Condition (3) in Theorem 1 is used to bound the residual term in normal approximation by the nonuniform Berry-Esseen Theorem [28]. Since the minimal value of the left hand side is $n^{-1/2}$ (corresponding to a WNN classifier where every data point has an equal vote of $1/n$,) this condition suggests that $n^{-1/2} = o(s^{-1/2}(\log(s))^{-2})$, i.e., roughly speaking, $s/n = o(1)$, or $\gamma < 1/2$. When $kNN$ is trained on each subsample, the condition reduces to $k^{-1/2} = o(s^{-1/2}(\log(s))^{-2})$ which means $s$ has a smaller order than the number of effective nearest neighbors $k$ on each subsample.*

From [46], we know that the minimal asymptotic regret of the oracle $K$NN ('oracle' means the classifier is obtained directly using the entire data set; we use $K$ to denote the number of neighbors here to indicate its global nature) is achieved when

$$K = K^* := \Big(\frac{dB_1}{4B_2}\Big)^{d/(d+4)} N^{4/(d+4)}.$$

Consider a special case for M-DiNN in which $kNN$ is trained at each local subsample, dubbed as M-DiNN($k$). An intuitive choice for $k$, the number of local neighbors for each local $kNN$ classifier, here is $\lceil K^*/s\rceil$, so that globally about $K^*$ neighbors are used. However, a direct application of Theorem 1 reveals that the optimal choice of $k$ for the M-DiNN($k$) method turns out to be

$$k^* = \lceil(\pi/2)^{d/(d+4)}(K^*/s)\rceil, \tag{5}$$

which is different from the intuitive choice above. The factor $\pi/2$ in (4) has led to the scaling constant $(\pi/2)^{d/(d+4)} > 1$ here in (5), which depends on the dimension only.

Given any general weight vector, Theorem 2 allows an asymptotic regret comparison between the M-DiNN and the oracle WNN, as implied by Proposition 1 and Theorem 1. Note that they share the same convergence rate and their difference is at the constant level.

**Theorem 2** *(Asymptotic Regret Comparison between M-DiNN and Oracle WNN) Suppose the conditions in Theorem 1 hold. Given an oracle WNN classifier with weights $\boldsymbol{w}_N$, denoted as $\widehat{\phi}_{N,\boldsymbol{w}_N}(x)$, there exists a M-DiNN classifier with weight $\boldsymbol{w}_n$, so that as $n, s \to \infty$,*

$$\frac{\mathrm{Regret}(\widehat{\phi}^M_{n,s,\boldsymbol{w}_n})}{\mathrm{Regret}(\widehat{\phi}_{N,\boldsymbol{w}_N})} \longrightarrow Q := \Big(\frac{\pi}{2}\Big)^{\frac{4}{d+4}},$$

*uniformly for $\boldsymbol{w}_n \in W_{n,\beta}$ and $\boldsymbol{w}_N \in W_{N,\beta}$ satisfying*

$$\frac{1}{s}\sum_{i=1}^{n} w_{ni}^2 / \sum_{i=1}^{N} w_{Ni}^2 \longrightarrow \Big(\frac{\pi}{2}\Big)^{-\frac{d}{d+4}} \quad \text{and} \tag{6}$$

$$\sum_{i=1}^{n} \frac{\alpha_i w_{ni}}{n^{2/d}} / \sum_{i=1}^{N} \frac{\alpha_i w_{Ni}}{N^{2/d}} \longrightarrow \Big(\frac{\pi}{2}\Big)^{\frac{2}{d+4}}. \tag{7}$$

Theorem 2 suggests if the local weights for M-DiNN are chosen to align with the oracle global weights according to (6) and (7), then M-DiNN and the oracle WNN achieve the same regret convergence rate. The ratio of these regrets is a constant, denoted as $Q$, which only depends on the data dimension. In Figure 1, we see that $Q$ is always smaller than 1.5, and monotonically decreases to 1 as $d$ grows. This may be viewed as a kind of "blessing of dimensionality." As will be shown, $Q$ results from the majority voting step in Algorithm 1, and thus we name it as the majority voting (MV) constant from now on.

As an illustration, we show how to find the local weights by applying the results in Theorem 2 to the OWNN method, which is the best oracle WNN method due to [46], whose global weights are defined as

$$w_i^*(N, m^*) = \begin{cases} \frac{1}{m^*}\Big[1 + \frac{d}{2} - \frac{d\alpha_i}{2(m^*)^{2/d}}\Big], & \text{if } i = 1, \ldots, m^*, \\ 0, & \text{if } i = m^* + 1, \ldots, N, \end{cases} \tag{8}$$

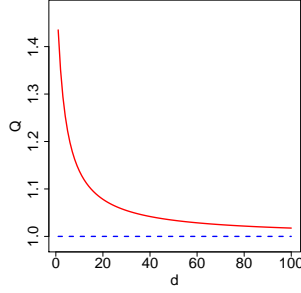

Figure 1: $Q$ for different $d$.

where

$$m^* = \lceil \left\{ \frac{d(d+4)}{2(d+2)} \right\}^{\frac{d}{d+4}} \left( \frac{B_1}{B_2} \right)^{\frac{d}{d+4}} N^{\frac{4}{d+4}} \rceil.$$

According to (6) and (7), the local weights in the optimal M-DiNN should be assigned as $w_{ni}^* := w_i^*(n, l^*)$, where

$$l^* = \lceil (\pi/2)^{d/(d+4)} (m^*/s) \rceil. \tag{9}$$

Interestingly, the above scaling factor is the same as that in (5) which relates the optimal numbers of neighbors for M-DiNN($k$) and the oracle optimal $K$NN.

Corollary 1 summarizes the above findings, and further discovers, in statement (ii), that

$$s^* \asymp N^{2/(d+4)}$$

is a *sharp* upper bound for the number of partitions in order for the M-DiNN method with optimal weights to achieve the same regret rate of the oracle OWNN. The ratio between the regrets is the same multiplicative constant $Q$ as stipulated in Theorem 2.

**Corollary 1** *(Optimal M-DiNN) Suppose the conditions in Theorem 1 hold.*

*(i) If $\gamma < 2/(d+4)$, the asymptotic minimum regret of M-DiNN is achieved by setting $w_{ni}^* = w_i^*(n, l^*)$ with $l^*$ defined in (9) and $w_i^*(\cdot, \cdot)$ defined in (8). In addition, we have as $n, s \to \infty$,*

$$\frac{\text{Regret}(\widehat{\phi}_{n,s,\boldsymbol{w}_n^*}^M)}{\text{Regret}(\widehat{\phi}_{N,\boldsymbol{w}_N^*})} \longrightarrow Q. \tag{10}$$

*(ii) If $\gamma \geq 2/(d+4)$, we have uniformly for $\boldsymbol{w}_n \in W_{n,\beta}$,*

$$\liminf_{n,s \to \infty} \frac{\text{Regret}(\widehat{\phi}_{n,s,\boldsymbol{w}_n}^M)}{\text{Regret}(\widehat{\phi}_{N,\boldsymbol{w}_N^*})} \longrightarrow \infty.$$

The upper bound on $s$ (or $\gamma$) in (ii) makes sense, as if there are too many partitions, there will be too few observations in each subsample, and the bias would be too large.

## 4 DiNN classifier via weighted voting

In this section, we investigate another type of distributed WNN based on weighted voting, denoted as W-DiNN. As will be shown, the weighted voting scheme helps to eliminate the multiplicative constant $Q$ in M-DiNN which represents a loss in regret compared to the oracle method. Specifically, the local classifier $\widehat{\phi}_{n,\boldsymbol{w}_n}^{(j)}(x)$ in Algorithm 1, which outputs 0 or 1, is replaced by the regression function estimator $\widehat{S}_{n,\boldsymbol{w}_n}^{(j)}(x) := \sum_{i=1}^n w_{ni} Y_{(i)}^{(j)}$, which outputs a number $\in [0, 1]$ (as opposed to $\{0, 1\}$ in M-DiNN.) The resulting classifier is defined as

$$\widehat{\phi}_{n,s,\boldsymbol{w}_n}^W(x) := \mathbb{1} \left\{ \frac{1}{s} \sum_{j=1}^s \widehat{S}_{n,\boldsymbol{w}_n}^{(j)}(x) \geq 1/2 \right\}.$$

The superscript $M$ in all notations used in Section 3 will be replaced by $W$ in this section.

The above simple change leads to a different asymptotic expansion of regret for W-DiNN, as stated in Theorem 3. Specifically, the variance term is reduced by $1/s$ in contrast to $\pi/(2s)$ for M-DiNN. Additionally, Condition (3) in Theorem 1 is no longer required in Theorem 3.

**Theorem 3** *(Asymptotic Regret for W-DiNN) Assuming the same conditions as in Proposition 1, we have for each $\beta \in (0, 1/2)$, as $n, s \to \infty$,*

$$\text{Regret}(\widehat{\phi}^W_{n,s,\boldsymbol{w}_n}) = \Big[ B_1 \frac{1}{s} \sum_{i=1}^n w_{ni}^2 + B_2 \Big( \sum_{i=1}^n \frac{\alpha_i w_{ni}}{n^{2/d}} \Big)^2 \Big] \{1 + o(1)\}, \tag{11}$$

*uniformly for $\boldsymbol{w}_n \in W_{n,\beta}$.*

A consequence of this new asymptotic expansion is that the optimal local choice of $k$ in W-DiNN($k$) (which gives rise to the same regret as the optimal oracle $K$NN) is indeed the intuitive choice of $k^\dagger = \lceil K^*/s \rceil$. This is different from (5) in M-DiNN($k$).

Similar result to Theorem 2 can be obtained for W-DiNN. Specifically, Theorem 4 says that given an oracle WNN with a given weight vector, one can find a W-DiNN with matching regret convergence rate. However, what's encouraging now is that this can be done *without* incurring any loss in the regret, whether on the rate level or on the multiplicative constant level.

**Theorem 4** *(Asymptotic Regret Comparison between W-DiNN and Oracle WNN) Suppose the conditions in Theorem 3 hold. We have as $n, s \to \infty$,*

$$\frac{\text{Regret}(\widehat{\phi}^W_{n,s,\boldsymbol{w}_n})}{\text{Regret}(\widehat{\phi}_{N,\boldsymbol{w}_N})} \longrightarrow 1,$$

*uniformly for $\boldsymbol{w}_n \in W_{n,\beta}$ and $\boldsymbol{w}_N \in W_{N,\beta}$ satisfying*

$$\frac{1}{s} \sum_{i=1}^n w_{ni}^2 / \sum_{i=1}^N w_{Ni}^2 \longrightarrow 1 \text{ and} \tag{12}$$

$$\sum_{i=1}^n \frac{\alpha_i w_{ni}}{n^{2/d}} / \sum_{i=1}^N \frac{\alpha_i w_{Ni}}{N^{2/d}} \longrightarrow 1. \tag{13}$$

Applying Theorem 4 to OWNN we can identify the optimal local weights for W-DiNN (that can achieve the same regret as OWNN) as $w^\dagger_{ni} := w^*_i(n, l^\dagger)$, where

$$l^\dagger = \lceil m^*/s \rceil \tag{14}$$

and $w^*_i(\cdot, \cdot)$ is defined in (8). Interestingly, due to fewer assumptions made for W-DiNN, Corollary 2 obtains a *higher* sharp upper bound

$$s^\dagger \asymp N^{4/(d+4)}$$

on the number of subsamples, with which the above matching regret is achievable by the W-DiNN classifier, compared to the case in M-DiNN. This suggest that more subsamples can be dispatched to more machines to compute in parallel in the W-DiNN framework.

**Corollary 2** *(Optimal W-DiNN) Suppose the conditions in Theorem 3 hold.*

*(i) If $\gamma < 4/(d+4)$, the asymptotic minimum regret of W-DiNN is achieved by setting $w^\dagger_{ni} = w^*_i(n, l^\dagger)$ and $w^*_i(\cdot, \cdot)$ defined in (8). Additionally, we have as $n, s \to \infty$,*

$$\frac{\text{Regret}(\widehat{\phi}^W_{n,s,\boldsymbol{w}^\dagger_n})}{\text{Regret}(\widehat{\phi}_{N,\boldsymbol{w}^*_N})} \longrightarrow 1. \tag{15}$$

*(ii) If $\gamma \geq 4/(d+4)$, we have uniformly for $\boldsymbol{w}_n \in W_{n,\beta}$,*

$$\liminf_{n,s\to\infty} \frac{\text{Regret}(\widehat{\phi}^W_{n,s,\boldsymbol{w}_n})}{\text{Regret}(\widehat{\phi}_{N,\boldsymbol{w}^*_N})} \longrightarrow \infty.$$

# 5 Numerical studies

In this section, we empirically check the accuracy of the M-DiNN(k) and W-DiNN(k) methods compared with four benchmark methods: the oracle $K$NN and oracle OWNN methods as well as two Fast Approximate Nearest Neighbor search methods (FANN), which are based on k-d tree and cover tree. In Section 3 of Supplement, we illustrate the effectiveness and statistical guarantees of the DiNN methods using simulations.

All numerical studies are conducted on HPC clusters with two 12-core Intel Xeon Gold Skylake processors and four 10-core Xeon-E5 processors, with memory between 64 and 128 GB.

We have retained benchmark data sets HTRU2 [40], Gisette [29], Musk1 [20], Musk2 [21], Occupancy [9], Credit [54], and SUSY [2], from the UCI machine learning repository [38]. The test sample sizes are set as $\min(1000, \text{total sample size}/5)$. Parameters in the oracle $K$NN, OWNN and FANN are tuned using cross-validation, and the parameter $k$ in M-DiNN(k), W-DiNN(k) and parameter $l$ in M-DiNN, W-DiNN for each subsample are set using bridging formulas stated in our theorems. The empirical risk is calculated over 1000 replications.

In Table 1, we compare the empirical risk (test error) and the speedup factors of M-DiNN(k), W-DiNN(k), and FANN relative to oracle $K$NN. The speedup factor is defined as the computing time of the oracle $K$NN divided by the time of the slower one between the two DiNN(k) methods or that of either FANN method. Because OWNN typically has a similar computing time as oracle $K$NN, the speed comparison with OWNN is omitted. From Table 1, we can see that the W-DiNN(k) has a similar risk as the oracle $K$NN, while the M-DiNN(k) has a somewhat larger risk. Both DiNN methods have a little larger risks compared with the oracle OWNN method, which has an optimally chosen weight function. Both DiNN methods also have significant smaller risks than the two FANN methods. In addition, the DiNN methods have a computational advantage over oracle $K$NN and oracle OWNN, and this advantage increases as the overall sample size increases (for a given $\gamma$). It seems that larger $\gamma$ values may induce slightly worse performance for the DiNN(k) classifiers, although such an observation is not conclusive. In addition, when $\gamma = 0.2, 0.3$, the DiNN methods have a larger speedup effect than the FANN methods.

Table 1: Risk (in %) of M-DiNN(k) (M($k$)) and W-DiNN(k) (W($k$)) compared to Fast Approximate Nearest Neighbor Search (FANN) (k-d tree (KT), cover tree (CT)), oracle $K$NN and OWNN. The speedup factors (SU-Di, SU-KT, SU-CT) are defined as the computing time of the oracle $K$NN divided by the time of the slower of the two DiNN(k) methods, and the two FANN methods, respectively.

| Data | N | d | $\gamma$ | M($k$) | W($k$) | KT | CT | $k$NN | OWNN | SU-Di | SU-KT | SU-CT |
|---|---|---|---|---|---|---|---|---|---|---|---|---|
| | | | 0.1 | 15.36 | 15.22 | | | | | 1.19 | | |
| Musk1 | 476 | 166 | 0.2 | 15.45 | 15.28 | 23.20 | 23.43 | 15.10 | 14.98 | 2.23 | 2.31 | 2.03 |
| | | | 0.3 | 15.82 | 15.53 | | | | | 3.30 | | |
| | | | 0.1 | 4.01 | 3.70 | | | | | 2.54 | | |
| Gisette | 6000 | 5000 | 0.2 | 4.18 | 3.94 | 7.11 | 7.16 | 3.62 | 3.48 | 4.55 | 1.56 | 1.13 |
| | | | 0.3 | 4.10 | 3.88 | | | | | 10.68 | | |
| | | | 0.1 | 3.91 | 3.78 | | | | | 3.30 | | |
| Musk2 | 6598 | 166 | 0.2 | 3.91 | 3.75 | 6.17 | 6.14 | 3.54 | 3.28 | 5.69 | 3.67 | 1.9 |
| | | | 0.3 | 4.23 | 3.98 | | | | | 15.62 | | |
| | | | 0.1 | 2.26 | 2.20 | | | | | 3.27 | | |
| HTRU2 | 17898 | 8 | 0.2 | 2.23 | 2.18 | 2.35 | 2.37 | 2.19 | 3.12 | 7.96 | 7.35 | 2.12 |
| | | | 0.3 | 2.30 | 2.22 | | | | | 21.90 | | |
| | | | 0.1 | 0.69 | 0.65 | | | | | 3.01 | | |
| Occup | 20560 | 6 | 0.2 | 0.75 | 0.73 | 0.83 | 0.87 | 0.65 | 0.60 | 7.47 | 10.02 | 6.78 |
| | | | 0.3 | 0.86 | 0.80 | | | | | 21.25 | | |
| | | | 0.1 | 19.37 | 19.28 | | | | | 3.00 | | |
| Credit | 30000 | 24 | 0.2 | 19.31 | 19.23 | 22.78 | 22.77 | 19.08 | 18.96 | 7.67 | 7.53 | 3.36 |
| | | | 0.3 | 19.33 | 19.27 | | | | | 23.57 | | |
| | | | 0.1 | 23.58 | 22.32 | | | | | 4.02 | | |
| SUSY | 5000K | 18 | 0.2 | 23.63 | 22.30 | 28.02 | 28.35 | 21.57 | 21.11 | 16.56 | 8.02 | 3.25 |
| | | | 0.3 | 23.76 | 22.51 | | | | | 72.78 | | |

# 6 Discussions

There are a couple of interesting directions to be pursued in the future. The first two are extensions to the multicategory classification problem and to high-dimensional data. The third direction is related to a realistic attack paradigm named adversarial examples that received a lot of recent attentions [48, 44]. [50] proposed a theoretical framework for learning robustness to adversarial examples and introduced a modified 1-nearest neighbor algorithm with good robustness. [53, 4] further studied the statistical guarantees for the robustness of nearest neighbor classifiers. It leaves us wonder how to take advantage of the distributed nature of DiNN to deal with adversarial samples.

Our main results assume bounded support for $X$, which follows that of [46] in which the minimax rate was attained with smoonthness assumption and margin condition as stated in [49], along with bounded support and the strong density condition. The latter two conditions may be too strong and exclude some common distributions. [25] showed that a relaxation to the strong minimal mass assumption and the tail assumption leads to suboptimal rate of convergence, while adaptive choice of $k$ can attain the minimax rate up to a logarithmic factor. A semi-supervised setting for this problem was discussed in [10], which considered a moment condition with more smoothness assumed near the boundary and the strong minimal mass assumption elsewhere. [22] used a modified Lipschitz condition or a local Lipschitz condition coupled with a weak density condition. We could consider these other assumptions to alleviate the bounded support assumption as future work, but choose not to do so in the current work to avoid a distraction from our main message.

## Broader Impact

Our work provides a statistical guarantee for the performance of nearest neighbor classifiers under the distributed framework. It also provides a guideline to choose suitable number of machines for parallel computing in practice. In addition, this paper reveals the accuracy lost of the majority voting scheme compared to the weighted voting scheme, which sheds light upon the choice of the voting scheme. Our work will advance the theoretical frontier for the nearest neighbor classifier.

## Acknowledgments

Guang Cheng was a member of the Institute for Advanced Study in writing this paper and would like to thank the institute for its hospitality.

## Footnotes

[1]When computing is done in parallel among all subsamples, the space complexity of M-DiNN is $dn$ at each subsample, much lower than $dN$, and the time complexity is reduced to $n \log(n)$ from $N \log(N)$

[2]In the case of $k$NN, it means $k$ satisfies $\max(n^\beta, (\log n)^2) \leq k \leq \min(n^{(1-\beta d/4)}, n^{1-\beta})$.

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
