[Supplementary Material]

# Supplement to: Statistical Guarantees of Distributed Nearest Neighbor Classification

**Jiexin Duan**
Department of Statistics
Purdue University
West Lafayette, Indiana, USA
duan32@purdue.edu

**Xingye Qiao**
Department of Mathematical Sciences
Binghamton University
New York, USA
qiao@math.binghamton.edu

**Guang Cheng**
Department of Statistics
Purdue University
West Lafayette, Indiana, USA
chengg@purdue.edu

## 1   Appendix 1: Assumptions (A1) - (A4)

For a smooth function $g$, we write $\dot{g}(x)$ for its gradient vector at $x$. The following conditions are assumed throughout this paper.

(A1) The set $\mathcal{R} \subset \mathbb{R}^d$ is a compact $d$-dimensional manifold with boundary $\partial \mathcal{R}$.

(A2) The set $\mathcal{S} = \{x \in \mathcal{R} : \eta(x) = 1/2\}$ is nonempty. There exists an open subset $U_0$ of $\mathbb{R}^d$ which contains $\mathcal{S}$ such that: (1) $\eta$ is continuous on $U \backslash U_0$ with $U$ an open set containing $\mathcal{R}$; (2) the restriction of the conditional distributions of $X$, $P_1$ and $P_0$, to $U_0$ are absolutely continuous with respect to Lebesgue measure, with twice continuously differentiable Randon-Nikodym derivatives $f_1$ and $f_0$.

(A3) There exists $\rho > 0$ such that $\int_{\mathbb{R}^d} \|x\|^\rho d\bar{P}(x) < \infty$. In addition, for sufficiently small $\delta > 0$, $\inf_{x \in \mathcal{R}} \bar{P}(B_\delta(x))/(a_d \delta^d) \ge C_0 > 0$, where $a_d = \pi^{d/2}/\Gamma(1 + d/2)$, $\Gamma(\cdot)$ is gamma function, and $C_0$ is a constant independent of $\delta$.

(A4) For all $x \in \mathcal{S}$, we have $\dot{\eta}(x) \ne 0$, and for all $x \in \mathcal{S} \cap \partial \mathcal{R}$, we have $\dot{\partial \eta}(x) \ne 0$, where $\partial \eta$ is the restriction of $\eta$ to $\partial \mathcal{R}$.  ∎

## 2   Appendix 2: Definitions of $a(x)$, $B_1$, $B_2$, $W_{n,\beta}$ and $W_{N,\beta}$

For a smooth function $g: \mathbb{R}^d \to \mathbb{R}$, denote $g_j(x)$ as its $j$-th partial derivative at $x$ and $g_{jk}(x)$ the $(j, k)$-th element of its Hessian matrix at $x$. Let $c_{j,d} = \int_{v:\|v\|\le 1} v_j^2 dv$, $\bar{f} = \pi_1 f_1 + (1 - \pi_1) f_0$. Define

$$a(x) = \sum_{j=1}^{d} \frac{c_{j,d}\{\eta_j(x)\bar{f}_j(x) + 1/2\eta_{jj}(x)\bar{f}(x)\}}{a_d^{1+2/d}\bar{f}(x)^{1+2/d}}.$$

Moreover, define two distribution-related constants

$$B_1 = \int_{\mathcal{S}} \frac{\bar{f}(x)}{4\|\dot{\eta}(x)\|}d\mathrm{Vol}^{d-1}(x), \quad B_2 = \int_{\mathcal{S}} \frac{\bar{f}(x)}{\|\dot{\eta}(x)\|}a(x)^2 d\mathrm{Vol}^{d-1}(x),$$

where $\mathrm{Vol}^{d-1}$ is the natural $(d-1)$-dimensional volume measure that $\mathcal{S}$ inherits as a subset of $\mathbb{R}^d$. According to Assumptions (A1)-(A4) in Appendix 1, $B_1$ and $B_2$ are finite with $B_1 > 0$ and $B_2 \geq 0$, with equality only when $a(x) = 0$ on $\mathcal{S}$.

In addition, for $\beta > 0$, we define $W_{n,\beta}$ as the set of $\boldsymbol{w}_n$ satisfying:

(w.1) $\sum_{i=1}^n w_{ni}^2 \leq n^{-\beta}$;

(w.2) $n^{-4/d}(\sum_{i=1}^n \alpha_i w_{ni})^2 \leq n^{-\beta}$, where $\alpha_i = i^{1+\frac{2}{d}} - (i-1)^{1+\frac{2}{d}}$;

(w.3) $n^{2/d}\sum_{i=k_2+1}^n w_{ni}/\sum_{i=1}^n \alpha_i w_{ni} \leq 1/\log n$ with $k_2 = \lceil n^{1-\beta} \rceil$;

(w.4) $\sum_{i=k_2+1}^n w_{ni}^2/\sum_{i=1}^n w_{ni}^2 \leq 1/\log n$;

(w.5) $\sum_{i=1}^n w_{ni}^3/(\sum_{i=1}^n w_{ni}^2)^{3/2} \leq 1/\log n$.

When $n$ in (w.1)–(w.5) is replaced by $N$, we can define the set $W_{N,\beta}$. ∎

## 3    Simulations

In this section, we compare various DiNN methods with the oracle $K$NN and the oracle OWNN methods respectively, with slightly different emphases. In comparing DiNN($k$) ($k$NN is trained at each subsample) with the oracle $K$NN, we aim to verify the main results in Theorem 2 and Theorem 4, namely, the M-DiNN and W-DiNN can approximate or attain the same performance as the oracle method. In comparing the DiNN methods with optimal local weights and the oracle OWNN method, we aim to verify the sharpness of upper bound on $\gamma$ in Corollary 1 and 2. This is by showing that the difference in performance between the DiNN methods and the oracle OWNN deviates when $\gamma$ is greater than the theoretical upper bound identified in these results.

Three simulation settings are considered. Simulation 1 allows a relatively easy classification task, Simulation 2 examines the bimodal effect, and Simulation 3 combines bimodality with dependence between variables.

In Simulation 1, $N = 27000$ and $d = 4, 6, 8$. The two classes are generated as $P_1 \sim N(0_d, \mathbb{I}_d)$ and $P_0 \sim N(\frac{2}{\sqrt{d}}1_d, \mathbb{I}_d)$ with the prior class probability $\pi_1 = \mathbb{P}(Y=1) = 1/3$. Simulation 2 has the same setting as Simulation 1, except that both classes are bimodal with $P_1 \sim 0.5N(0_d, \mathbb{I}_d) + 0.5N(3_d, 2\mathbb{I}_d)$ and $P_0 \sim 0.5N(1.5_d, \mathbb{I}_d) + 0.5N(4.5_d, 2\mathbb{I}_d)$. Simulation 3 has the same setting as Simulation 2, except that $P_1 \sim 0.5N(0_d, \Sigma) + 0.5N(3_d, 2\Sigma)$ and $P_0 \sim 0.5N(1.5_d, \Sigma) + 0.5N(4.5_d, 2\Sigma)$ with $\pi_1 = 1/2$, where $\Sigma$ is the Toeplitz matrix whose $j$-th entry of the first row is $0.6^{j-1}$.

Recall $s = N^\gamma$. We let the exponent $\gamma = 0.0, 0.1 \ldots 0.8$. When comparing the $k$NN methods, the number of neighbors $K$ in the oracle $K$NN is chosen as $K = N^{0.7}$. The number of local neighbors in M-DiNN($k$) and W-DiNN($k$) are chosen as $k = \lceil (\pi/2)^{d/(d+4)}K/s \rceil$ and $k = \lceil K/s \rceil$ as suggested by Theorem 2 and Theorem 4 respectively. These $k$ values are truncated at 1, since we cannot have a fraction of an observation. In comparing with the oracle OWNN method, the $m^*$ parameter in OWNN is tuned using cross-validation. The parameter $l$ in M-DiNN and W-DiNN for each subsample are chosen as $l^* = \lceil (\pi/2)^{d/(d+4)}(m^*/s) \rceil$ and $l^\dagger = \lceil m^*/s \rceil$ as stated in Corollary 1 and Corollary 2 respectively. For both comparisons, the test set is independently generated with 1000 observations. We repeat the simulation for 1000 times for each $\gamma$ and $d$. Here the empirical risk (test error) and the computation time are calculated for each of the methods.

Figure S1 shows that M-DiNN($k$) and W-DiNN($k$) require similar computing time, and both are significantly faster than the oracle method. As the number of subsamples increases, the running time decreases, which shows the time benefit of the distributed learning framework. The computing time comparison with the oracle OWNN is omitted since the message is the same.

The comparison between the risks of the three $k$NN methods (one oracle and two distributed) are reported in Figure S2. For smaller $\gamma$ values, the risk curve for W-DiNN($k$) overlaps with that of the oracle kNN, while the curve for M-DiNN($k$) has a conceivable gap with both. These verify the main results in Theorem 2 and Theorem 4. The performance of the M-DiNN($k$) method starts to deviate from the oracle kNN since $\gamma = 0.6$. As $s$ increases and goes beyond the threshold $\gamma = 0.7$, the risk deteriorates more quickly. These may be caused by the finite (or very small) number of voting neighbors $k$ at each subsample, which means the requirements $k = \lceil (\pi/2)^{d/(d+4)}K/s \rceil \to \infty$,

Figure S1: Computation time (seconds) of M-DiNN($k$), W-DiNN($k$), and oracle $K$NN for different $\gamma$. Left/middle/right: Simulation $1/2/3$, $d = 4/6/8$.

Figure S2: Risk of M-DiNN($k$), W-DiNN($k$), oracle $K$NN and the Bayes rule for different $\gamma$. Top/middle/bottom: Simulation $1/2/3$; left/middle/right: $d = 4/6/8$. $\gamma = 0.7$ is shown as a vertical line: DiNN methods at or after this line have only 1 nearest neighbor at each subsample which participates in the prediction.

$k = \lceil K/s \rceil \to \infty$ suggested by Theorem 2 and Theorem 4 respectively are not satisfied. Specifically, when $\gamma = 0.6, 0.7, 0.8$, the number of voting neighbors $k$ are no more than 3 in these simulated examples. We did not tune the parameters and simply set $K$ in the oracle $K$NN as $N^{0.7}$, since the results in Theorem 2 and Theorem 4 should hold for any reasonable weights (or reasonable choice of $k$), not necessarily the optimal one.

On the other hand, since the comparison with the oracle OWNN is meant to verify the sharp upper bound for $\gamma$ in the optimal weight setting (Corollary 1 and Corollary 2), we carefully tune the weights in the oracle OWNN method in order to reach the optimality. Figure S3 shows the comparison of risks for M-DiNN, W-DiNN and oracle OWNN methods. Our focus here is when the two DiNN

Figure S3: Risk of optimal M-DiNN, W-DiNN, oracle OWNN and the Bayes rule for different $\gamma$. Left/middle/right: Simulation $1/2/3$, $d = 4$. Upper bounds for number of subsamples in optimal M-DiNN ($\gamma = 1/4$) and W-DiNN ($\gamma = 1/2$) are shown as two vertical lines.

methods start to have significantly worse performance than the oracle OWNN, and the answers lie in the upper bounds in Corollary 1 and Corollary 2. For simplicity, we set $d = 4$, which leads to an upper bound of $2/(d+4) = 0.25$ for the M-DiNN method, and an upper bound of $4/(d+4) = 0.5$ for the W-DiNN method. These upper bounds are shown as vertical lines in Figure S3. Specifically, the M-DiNN deteriorates much earlier than W-DiNN with much few machines (subsamples) at its disposal. The W-DiNN method performs much better than M-DiNN, having almost the same performance as the OWNN method for $\gamma \leq 0.4$. However, even W-DiNN does not perform well enough for $\gamma \geq 4/(d+4) = 0.5$ when compared to OWNN. These verify the results in Corollary 2 and Corollary 4.

## 4  Proof of Theorem 1

For the sake of simplicity, we omit $\boldsymbol{w}_n$ in the subscript of such notations as $\widehat{\phi}_{n,s,\boldsymbol{w}_n}^{M}$ and $S_{n,\boldsymbol{w}_n}^{(j)}$. Write $P^{\circ} = \pi_1 P_1 - (1 - \pi_1) P_0$. We have

$$
\begin{aligned}
\text{Regret}(\widehat{\phi}_{n,s}^{M}) &= R(\widehat{\phi}_{n,s}^{M}) - R(\phi^*) \\
&= \int_{\mathcal{R}} \pi_1 \big[ \mathbb{P}\big( \widehat{\phi}_{n,s}^{M}(x) = 0 \big) - \mathbb{1}\big\{ \phi^*(x) = 0 \big\} \big] dP_1(x) \\
&\quad + \int_{\mathcal{R}} (1 - \pi_1) \big[ \mathbb{P}\big( \widehat{\phi}_{n,s}^{M}(x) = 1 \big) - \mathbb{1}\big\{ \phi^*(x) = 1 \big\} \big] dP_0(x) \\
&= \int_{\mathcal{R}} \big[ \mathbb{P}\big( \widehat{\phi}_{n,s}^{M}(x) = 0 \big) - \mathbb{1}\big\{ \eta(x) < 1/2 \big\} \big] dP^{\circ}(x).
\end{aligned}
$$

Without loss of generality, we consider the $j$-th subsample of $\mathcal{D}$: $\mathcal{D}^{(j)} = \{(X_i^{(j)}, Y_i^{(j)}), i = 1, \ldots, n\}$. Given $X = x$, we define $(X_{(i)}^{(j)}, Y_{(i)}^{(j)})$ such that $\|X_{(1)}^{(j)} - x\| \leq \|X_{(2)}^{(j)} - x\| \leq \ldots \leq \|X_{(n)}^{(j)} - x\|$. Denote the estimated regression function on the $j$-th subsample as

$$
S_n^{(j)}(x) = \sum_{i=1}^{n} w_{ni} Y_{(i)}^{(j)}.
$$

Denote the WNN classifier on the $j$-th subsample as

$$
\widehat{\phi}_n^{(j)}(x) = \mathbb{1}\big\{ S_n^{(j)}(x) \geq 1/2 \big\}.
$$

For any $j$ and $x$, we have $\mathbb{P}\big( S_n^{(j)}(x) \geq 1/2 \big) = \mathbb{P}\big( S_n(x) \geq 1/2 \big)$, where $S_n(x)$ is a generic local WNN regression function on any subsample. Hence, $\widehat{\phi}_n^{(j)}(x)$ $(j = 1, \ldots, s)$ have i.i.d. Bernoulli distribution with success probability $\mathbb{P}\big( S_n(x) \geq 1/2 \big)$. In particular, we have

$$
\begin{aligned}
\mathbb{E}\{\widehat{\phi}_n^{(j)}(x)\} &= \mathbb{P}\big( S_n(x) \geq 1/2 \big), \\
Var\{\widehat{\phi}_n^{(j)}(x)\} &= \mathbb{P}\big( S_n(x) < 1/2 \big) \mathbb{P}\big( S_n(x) \geq 1/2 \big).
\end{aligned}
$$

Denote the average of the predictions from $s$ subsamples as

$$
S_{n,s}^{M}(x) = s^{-1} \sum_{j=1}^{s} \widehat{\phi}_n^{(j)}(x).
$$

Therefore,

$$\mathbb{E}\{S_{n,s}^M(x)\} = \mathbb{P}\big(S_n(x) \geq 1/2\big),$$
$$Var\{S_{n,s}^M(x)\} = s^{-1}\mathbb{P}\big(S_n(x) < 1/2\big)\mathbb{P}\big(S_n(x) \geq 1/2\big).$$

The M-DiNN classifier is defined as

$$\widehat{\phi}_{n,s}^M(x) = \mathbb{1}\big\{S_{n,s}^M(x) \geq 1/2\big\}.$$

Since $\mathbb{P}\big(\widehat{\phi}_{n,s}^M(x) = 0\big) = \mathbb{P}\big(S_{n,s}^M(x) < 1/2\big)$, the regret of M-DiNN becomes

$$\text{Regret}(\widehat{\phi}_{n,s}^M) = \int_{\mathcal{R}} \big\{\mathbb{P}\big(S_{n,s}^M(x) < 1/2\big) - \mathbb{1}\{\eta(x) < 1/2\}\big\}dP^\circ(x).$$

In any subsample, denote the boundary $\mathcal{S} = \{x \in \mathcal{R} : \eta(x) = 1/2\}$. For $\epsilon > 0$, let $\mathcal{S}^{\epsilon\epsilon} = \{x \in \mathbb{R}^d : \eta(x) = 1/2 \text{ and } \text{dist}(x, \mathcal{S}) < \epsilon\}$, where $\text{dist}(x, \mathcal{S}) = \inf_{x_0 \in \mathcal{S}} \|x - x_0\|$. We will focus on the set

$$\mathcal{S}^\epsilon = \Big\{x_0 + t\frac{\dot{\eta}(x_0)}{\|\dot{\eta}(x_0)\|} : x_0 \in \mathcal{S}^{\epsilon\epsilon}, |t| < \epsilon\Big\}.$$

Let $\mu_n(x) = \mathbb{E}\{S_n(x)\}$, $\sigma_n^2(x) = \text{Var}\{S_n(x)\}$, and $\epsilon_n = n^{-\beta/(4d)}$. Denote $s_n^2 = \sum_{i=1}^n w_{ni}^2$ and $t_n = n^{-2/d}\sum_{i=1}^n \alpha_i w_{ni}$. Samworth [5] showed that, uniformly for $\boldsymbol{w}_n \in W_{n,\beta}$,

$$\sup_{x \in \mathcal{S}^{\epsilon_n}} |\mu_n(x) - \eta(x) - a(x)t_n| = o(t_n), \tag{S.1}$$

$$\sup_{x \in \mathcal{S}^{\epsilon_n}} \big|\sigma_n^2(x) - \frac{1}{4}s_n^2\big| = o(s_n^2). \tag{S.2}$$

Let $\epsilon_{n,s}^M = a_0 t_n + b_0 \frac{\log(s)}{\sqrt{s}}s_n$, where $a_0$ and $b_0$ are constants that $a_0 > \frac{2|a(x_0)|}{\|\dot{\eta}(x_0)\|}$ and $b_0 > \frac{\sqrt{2\pi}}{\|\dot{\eta}(x_0)\|}$, for any $x_0 \in \mathcal{S}$.

We organize our proof in four steps. In *Step 1*, we decompose the integral over $\mathcal{R} \cap \mathcal{S}^{\epsilon_n}$ as an integral along $\mathcal{S}$ and an integral in the perpendicular direction; in *Step 2*, we bound the contribution to regret from $\mathcal{R}\backslash\mathcal{S}^{\epsilon_n}$; in *Step 3*, we bound the contribution to regret from $\mathcal{S}^{\epsilon_n}\backslash\mathcal{S}^{\epsilon_{n,s}^M}$; *Step 4* combines the results in previous steps and applies the normal approximation in $\mathcal{S}^{\epsilon_{n,s}^M}$ to yield the final conclusion.

*Step 1*: For $x_0 \in \mathcal{S}$ and $t \in \mathbb{R}$, denote $x_0^t = x_0 + t\dot{\eta}(x_0)/\|\dot{\eta}(x_0)\|$. Denote $\psi = \pi_1 f_1 - (1 - \pi_1)f_0$, $\bar{f} = \pi_1 f_1 + (1 - \pi_1)f_0$ as the Radon-Nikodym derivatives with respect to Lebesgue measure of the restriction of $P^\circ$ and $\bar{P}$ to $\mathcal{S}^{\epsilon_n}$ for large $n$ respectively.

Similar to Samworth [5], we consider a change of variable from $x$ to $x_0^t$. By the theory of integration on manifolds and Weyl's tube formula [2], we have, uniformly for $\boldsymbol{w}_n \in W_{n,\beta}$,

$$\int_{\mathcal{R}\cap\mathcal{S}^{\epsilon_n}} \big\{\mathbb{P}\big(S_{n,s}^M(x) < 1/2\big) - \mathbb{1}\{\eta(x) < 1/2\}\big\}dP^\circ(x)$$
$$= \int_{\mathcal{S}} \int_{-\epsilon_n}^{\epsilon_n} \psi(x_0^t)\big\{\mathbb{P}\big(S_{n,s}^M(x_0^t) < 1/2\big) - \mathbb{1}\{t < 0\}\big\}dt d\text{Vol}^{d-1}(x_0)\{1 + o(1)\}.$$

*Step 2*: Bound the contribution to regret from $\mathcal{R}\backslash\mathcal{S}^{\epsilon_n}$. We show that,

$$\sup_{\boldsymbol{w}_n \in W_{n,\beta}} \int_{\mathcal{R}\backslash\mathcal{S}^{\epsilon_n}} \big\{\mathbb{P}\big(S_{n,s}^M(x) < 1/2\big) - \mathbb{1}\{\eta(x) < 1/2\}\big\}dP^\circ(x) = o(\frac{s_n^2}{s} + t_n^2).$$

According to [5], for all $M > 0$, uniformly for $\boldsymbol{w}_n \in W_{n,\beta}$ and $x \in \mathcal{R}\backslash\mathcal{S}^{\epsilon_n}$, we have

$$|\mathbb{P}\big(S_n(x) < 1/2\big) - \mathbb{1}\{\eta(x) < 1/2\}| = O(n^{-M}).$$

Therefore, for $x \in \mathcal{R}\backslash\mathcal{S}^{\epsilon_n}$ and sufficiently large $n$, we have

$$\inf_{\eta(x)<1/2} \mathbb{P}\big(S_n(x) < 1/2\big) - 1/2 > 1/4, \tag{S.3}$$

$$\sup_{\eta(x)\geq 1/2} \mathbb{P}\big(S_n(x) < 1/2\big) - 1/2 < -1/4. \tag{S.4}$$

Applying Hoeffding's inequality to $S_{n,s}^M(x)$, along with (S.3) and (S.4), we have

$$|\mathbb{P}(S_{n,s}^M(x) < 1/2) - \mathbb{1}\{\eta(x) < 1/2\}| \leq \exp\Big(\frac{-2(\mathbb{E}\{S_{n,s}^M(x)\} - 1/2)^2}{\sum_{j=1}^s (1/s - 0)^2}\Big)$$

$$= \exp\Big(\frac{-2(\mathbb{P}(S_n(x) < 1/2) - 1/2)^2}{1/s}\Big) = o(\frac{s_n^2}{s} + t_n^2),$$

uniformly for $\boldsymbol{w}_n \in W_{n,\beta}$ and $x \in \mathcal{R} \backslash \mathcal{S}^{\epsilon_n}$. This completes *Step 2*.

*Step 3*: Bound the contribution to regret from $\mathcal{S}^{\epsilon_n} \backslash \mathcal{S}^{\epsilon_{n,s}^M}$. We show that

$$\sup_{\boldsymbol{w}_n \in W_{n,\beta}} \int_{\mathcal{S}} \int_{(-\epsilon_n, \epsilon_n) \backslash (-\epsilon_{n,s}^M, \epsilon_{n,s}^M)} \psi(x_0^t)\{\mathbb{P}(S_{n,s}^M(x_0^t) < 1/2)$$
$$- \mathbb{1}\{t < 0\}\} dt d\mathrm{Vol}^{d-1}(x_0) = o(s_n^2/s + t_n^2).$$

For $x_0^t \in \mathcal{S}^{\epsilon_n} \backslash \mathcal{S}^{\epsilon_{n,s}^M}$, we have $t \notin (-\epsilon_{n,s}^M, \epsilon_{n,s}^M)$. By (S.1), (S.2) and Taylor expansion, we have

$$\frac{1/2 - \mu_n(x_0^t)}{\sigma_n(x_0^t)} = \frac{-t\|\dot{\eta}(x_0)\| - a(x_0)t_n + o(t_n)}{s_n/2 + o(s_n)}.$$

Since $t \notin (-\epsilon_{n,s}^M, \epsilon_{n,s}^M)$, $|t| \geq \epsilon_{n,s}^M = a_0 t_n + b_0 \frac{\log(s)}{\sqrt{s}} s_n$, we have for a sufficiently large n,

$$\Big|\frac{1/2 - \mu_n(x_0^t)}{\sigma_n(x_0^t)}\Big| > \sqrt{2\pi}\frac{\log(s)}{\sqrt{s}}.$$

If in addition, $\Big|\frac{1/2 - \mu_n(x_0^t)}{\sigma_n(x_0^t)}\Big| = o(1)$, then by Lemma 1, we have

$$\Big|\Phi\Big(\frac{1/2 - \mu_n(x_0^t)}{\sigma_n(x_0^t)}\Big) - 1/2\Big| > \frac{\log(s)}{\sqrt{s}},$$

where $\Phi$ is the standard normal distribution function.

Otherwise if $\Big|\frac{1/2 - \mu_n(x_0^t)}{\sigma_n(x_0^t)}\Big| > c_{10}$, where $c_{10}$ is a positive constant, then we have, for $s$ large enough,

$$\Big|\Phi\Big(\frac{1/2 - \mu_n(x_0^t)}{\sigma_n(x_0^t)}\Big) - 1/2\Big| > \frac{\log(s)}{\sqrt{s}}.$$

In summary, for $x_0^t \in \mathcal{S}^{\epsilon_n} \backslash \mathcal{S}^{\epsilon_{n,s}^M}$,

$$\Big|\Phi\Big(\frac{1/2 - \mu_n(x_0^t)}{\sigma_n(x_0^t)}\Big) - 1/2\Big| > \frac{\log(s)}{\sqrt{s}}. \tag{S.5}$$

Let $Z_i = (w_{ni}Y_{(i)} - w_{ni}\mathbb{E}[Y_{(i)}])/\sigma_n(x)$ and $W = \sum_{i=1}^n Z_i$. Note that $\mathbb{E}(Z_i) = 0$, $\mathrm{Var}(Z_i) < \infty$ and $\mathrm{Var}(W) = 1$. The nonuniform Berry-Esseen Theorem [3] implies that there exists a constant $c_{11} > 0$ such that

$$\Big|\mathbb{P}(W \leq By) - \Phi(y)\Big| \leq \frac{c_{11}A}{B^3(1 + |y|^3)},$$

where $A = \sum_{i=1}^n E|Z_i|^3$ and $B = (\sum_{i=1}^n E|Z_i|^2)^{1/2}$. In our case,

$$A = \sum_{i=1}^n \mathbb{E}|\frac{w_{ni}Y_i - w_{ni}\mathbb{E}[Y_i]}{\sigma_n(x)}|^3 \leq \sum_{i=1}^n \frac{2|w_{ni}|^3}{(s_n/2)^3} = \frac{16\sum_{i=1}^n w_{ni}^3}{(\sum_{i=1}^n w_{ni}^2)^{3/2}},$$

$$B = (\sum_{i=1}^n \mathrm{Var}(Z_i))^{1/2} = \sqrt{\mathrm{Var}(W)} = 1.$$

Let $c_{12} = 16c_{11}$, we have

$$\sup_{x_0 \in \mathcal{S}} \sup_{t \in [-\epsilon_n, \epsilon_n]} \Big|\mathbb{P}\Big(\frac{S_n(x_0^t) - \mu_n(x_0^t)}{\sigma_n(x_0^t)} \leq y\Big) - \Phi(y)\Big| \leq \frac{\sum_{i=1}^n w_{ni}^3}{(\sum_{i=1}^n w_{ni}^2)^{3/2}} \frac{c_{12}}{1 + |y|^3}.$$

Setting $y = \frac{1/2 - \mu_n(x_0^t)}{\sigma_n(x_0^t)}$, we have

$$\sup_{x_0 \in \mathcal{S}} \sup_{t \in [-\epsilon_n, \epsilon_n]} \left| \mathbb{P}\big(S_n(x_0^t) < 1/2\big) - \Phi\Big(\frac{1/2 - \mu_n(x_0^t)}{\sigma_n(x_0^t)}\Big) \right| \qquad \text{(S.6)}$$

$$\leq \frac{c_{12} \sum_{i=1}^n w_{ni}^3}{(\sum_{i=1}^n w_{ni}^2)^{3/2}} = o\Big(\frac{1}{\sqrt{s}(\log(s))^2}\Big).$$

The last equality holds by (3).

By (S.5) and (S.6), we have, when $x_0^t \in \mathcal{S}^{\epsilon_n} \backslash \mathcal{S}^{\epsilon_{n,s}^M}$,

$$\left| \mathbb{P}\big(S_n(x_0^t) < 1/2\big) - 1/2 \right| > \log(s)/\sqrt{s}. \qquad \text{(S.7)}$$

Applying Hoeffding's inequality to $S_{n,s}^M(x_0^t)$, we have

$$|\mathbb{P}(S_{n,s}^M(x_0^t) < 1/2) - \mathbb{1}\{t < 0\}| \leq \exp\Big[\frac{-2(\mathbb{E}\{S_{n,s}^M(x_0^t)\} - 1/2)^2}{\sum_{j=1}^s (1/s - 0)^2}\Big]$$

$$= \exp\big[-2s(\mathbb{P}(S_n(x_0^t) < 1/2) - 1/2)^2\big] < s^{-2\log(s)} = o(s_n^2/s + t_n^2),$$

uniformly for $\boldsymbol{w}_n \in W_{n,\beta}$ and $x_0^t \in \mathcal{S}^{\epsilon_n} \backslash \mathcal{S}^{\epsilon_{n,s}^M}$. This completes *Step 3*.

*Step 4*: In the end, we will show

$$\int_{\mathcal{S}} \int_{-\epsilon_{n,s}^M}^{\epsilon_{n,s}^M} \psi(x_0^t)\big\{\mathbb{P}\big(S_{n,s}^M(x_0^t) < 1/2\big) - \mathbb{1}\{t < 0\}\big\} dt d\text{Vol}^{d-1}(x_0)$$

$$= B_1 \frac{\pi}{2s} s_n^2 + B_2 t_n^2 + o\Big(\frac{s_n^2}{s} + t_n^2\Big).$$

Applying Taylor expansion, we have, for $x_0 \in \mathcal{S}$,

$$\psi(x_0^t) = \psi(x_0) + \dot{\psi}(x_0)^T (x_0^t - x_0) + o(x_0^t - x_0) \qquad \text{(S.8)}$$

$$= \dot{\psi}(x_0)^T \frac{\dot{\eta}(x_0)}{\|\dot{\eta}(x_0)\|} t + o(t)$$

$$= \|\dot{\psi}(x_0)\| t + o(t),$$

where the above second equality holds by definition of $x_0^t$, and the third equality holds by Lemma 4. Hence,

$$\int_{\mathcal{S}} \int_{-\epsilon_{n,s}^M}^{\epsilon_{n,s}^M} \psi(x_0^t)\big\{\mathbb{P}\big(S_{n,s}^M(x_0^t) < 1/2\big) - \mathbb{1}\{t < 0\}\big\} dt d\text{Vol}^{d-1}(x_0) \qquad \text{(S.9)}$$

$$= \int_{\mathcal{S}} \int_{-\epsilon_{n,s}^M}^{\epsilon_{n,s}^M} t\|\dot{\psi}(x_0)\|\big\{\mathbb{P}\big(S_{n,s}^M(x_0^t) < 1/2\big)$$

$$- \mathbb{1}\{t < 0\}\big\} dt d\text{Vol}^{d-1}(x_0)\{1 + o(1)\}.$$

Next, we decompose

$$\int_{\mathcal{S}} \int_{-\epsilon_{n,s}^M}^{\epsilon_{n,s}^M} t\|\dot{\psi}(x_0)\|\big\{\mathbb{P}\big(S_{n,s}^M(x_0^t) < 1/2\big) - \mathbb{1}\{t < 0\}\big\} dt d\text{Vol}^{d-1}(x_0) \qquad \text{(S.10)}$$

$$= \int_{\mathcal{S}} \int_{-\epsilon_{n,s}^M}^{\epsilon_{n,s}^M} t\|\dot{\psi}(x_0)\|\Big\{\Phi\Big[\frac{\sqrt{s}\big(1/2 - \mathbb{P}\big(S_n(x_0^t) \geq 1/2\big)\big)}{\sqrt{\mathbb{P}\big(S_n(x_0^t) < 1/2\big)\mathbb{P}\big(S_n(x_0^t) \geq 1/2\big)}}\Big]$$

$$- \mathbb{1}\{t < 0\}\Big\} dt d\text{Vol}^{d-1}(x_0) + R_{11}.$$

If $|1/2 - \mathbb{P}\big(S_n(x_0^t) < 1/2\big)| \leq \log(s)/\sqrt{s}$, by the uniform Berry-Esseen Theorem [4], there exists a constant $c_{13} > 0$ such that

$$\left|\mathbb{P}\Big(\frac{\sqrt{s}\big[S_{n,s}^M(x_0^t) - \mathbb{P}\big(S_n(x_0^t) \geq 1/2\big)\big]}{\sqrt{\mathbb{P}\big(S_n(x_0^t) < 1/2\big)\mathbb{P}\big(S_n(x_0^t) \geq 1/2\big)}} < y\Big) - \Phi(y)\right|$$

$$\leq \quad \frac{c_{13}}{\sqrt{s}} \frac{\mathbb{E}\big|\widehat{\phi}_n^{(j)}(x_0^t) - \mathbb{P}\big(S_n(x_0^t) \geq 1/2\big)\big|^3}{\big[\mathbb{P}\big(S_n(x_0^t) < 1/2\big)\mathbb{P}\big(S_n(x_0^t) \geq 1/2\big)\big]^{3/2}} \leq \frac{8c_{13}}{\sqrt{s}} = O\Big(\frac{1}{\sqrt{s}}\Big).$$

Setting $y = \frac{\sqrt{s}(1/2 - \mathbb{P}(S_n(x_0^t) \geq 1/2))}{\sqrt{\mathbb{P}(S_n(x_0^t) < 1/2)\mathbb{P}(S_n(x_0^t) \geq 1/2)}}$, we have

$$\left|\mathbb{P}\big(S_{n,s}^M(x_0^t) < 1/2\big) - \Phi\Big[\frac{\sqrt{s}\big(1/2 - \mathbb{P}\big(S_n(x_0^t) \geq 1/2\big)\big)}{\sqrt{\mathbb{P}\big(S_n(x_0^t) < 1/2\big)\mathbb{P}\big(S_n(x_0^t) \geq 1/2\big)}}\Big]\right| = O\Big(\frac{1}{\sqrt{s}}\Big).$$

In addition, if $|1/2 - \mathbb{P}\big(S_n(x_0^t) \geq 1/2\big)| > \frac{\log(s)}{\sqrt{s}}$, applying Hoeffding's inequality and Lemma 3, we have

$$|\mathbb{P}(S_{n,s}^M(x_0^t) < 1/2) - \mathbb{1}\big\{\mathbb{P}\big(S_n(x_0^t) \geq 1/2\big) < 1/2\big\}|$$

$$= \exp\big[-2s(1/2 - \mathbb{P}(S_n(x_0^t) \geq 1/2))^2\big] \leq \exp(-2[\log(s)]^2) = o\Big(\frac{1}{\sqrt{s}}\Big),$$

$$\left|\Phi\Big[\frac{\sqrt{s}\big(1/2 - \mathbb{P}\big(S_n(x_0^t) \geq 1/2\big)\big)}{\sqrt{\mathbb{P}\big(S_n(x_0^t) < 1/2\big)\mathbb{P}\big(S_n(x_0^t) \geq 1/2\big)}}\Big] - \mathbb{1}\big\{\mathbb{P}\big(S_n(x_0^t) \geq 1/2\big) < 1/2\big\}\right|$$

$$\leq \frac{1}{2\log(s)} \frac{e^{-[2\log(s)]^2/2}}{\sqrt{2\pi}} = o\Big(\frac{1}{\sqrt{s}}\Big).$$

In this case, we have

$$\left|\mathbb{P}\big(S_{n,s}^M(x_0^t) < 1/2\big) - \Phi\Big[\frac{\sqrt{s}\big(1/2 - \mathbb{P}\big(S_n(x_0^t) \geq 1/2\big)\big)}{\sqrt{\mathbb{P}\big(S_n(x_0^t) < 1/2\big)\mathbb{P}\big(S_n(x_0^t) \geq 1/2\big)}}\Big]\right|$$

$$\leq \left|\mathbb{P}\big(S_{n,s}^M(x_0^t) < 1/2\big) - \mathbb{1}\big\{\mathbb{P}\big(S_n(x_0^t) \geq 1/2\big) < 1/2\big\}\right|$$

$$+ \left|\Phi\Big[\frac{\sqrt{s}\big(1/2 - \mathbb{P}\big(S_n(x_0^t) \geq 1/2\big)\big)}{\sqrt{\mathbb{P}\big(S_n(x_0^t) < 1/2\big)\mathbb{P}\big(S_n(x_0^t) \geq 1/2\big)}}\Big] - \mathbb{1}\big\{\mathbb{P}\big(S_n(x_0^t) \geq 1/2\big) < \frac{1}{2}\big\}\right|$$

$$= o\Big(\frac{1}{\sqrt{s}}\Big).$$

In summary, we have

$$\sup_{x_0 \in \mathcal{S}} \sup_{t \in [-\epsilon_{n,s}^M, \epsilon_{n,s}^M]} \left|\mathbb{P}\big(S_{n,s}^M(x_0^t) < 1/2\big)\right. \qquad\qquad \text{(S.11)}$$

$$\left. - \Phi\Big[\frac{\sqrt{s}\big(1/2 - \mathbb{P}\big(S_n(x_0^t) \geq 1/2\big)\big)}{\sqrt{\mathbb{P}\big(S_n(x_0^t) < 1/2\big)\mathbb{P}\big(S_n(x_0^t) \geq 1/2\big)}}\Big]\right| = O\Big(\frac{1}{\sqrt{s}}\Big).$$

Thus, we have

$$|R_{11}| \leq \int_{\mathcal{S}} \int_{-\epsilon_{n,s}^M}^{\epsilon_{n,s}^M} |t| \|\dot{\psi}(x_0)\| \Big|\mathbb{P}\big(S_{n,s}^M(x_0^t) < 1/2\big)$$

$$- \Phi\Big[\frac{\sqrt{s}\big(1/2 - \mathbb{P}\big(S_n(x_0^t) \geq 1/2\big)\big)}{\sqrt{\mathbb{P}\big(S_n(x_0^t) < 1/2\big)\mathbb{P}\big(S_n(x_0^t) \geq 1/2\big)}}\Big]\Big| dt d\text{Vol}^{d-1}(x_0)$$

$$\leq O(\frac{1}{\sqrt{s}}) \int_{\mathcal{S}} \int_{-\epsilon_{n,s}^M}^{\epsilon_{n,s}^M} |t| \|\dot{\psi}(x_0)\| dt d\text{Vol}^{d-1}(x_0) = o(t_n^2 + \frac{1}{s}s_n^2).$$

Next, we decompose

$$\int_{\mathcal{S}} \int_{-\epsilon_{n,s}^M}^{\epsilon_{n,s}^M} t \|\dot{\psi}(x_0)\| \Big\{ \Phi\Big[\frac{\sqrt{s}\big(1/2 - \mathbb{P}\big(S_n(x_0^t) \geq 1/2\big)\big)}{\sqrt{\mathbb{P}\big(S_n(x_0^t) < 1/2\big)\mathbb{P}\big(S_n(x_0^t) \geq 1/2\big)}}\Big] \tag{S.12}$$
$$- \mathbb{1}\{t < 0\} \Big\} dt d\mathrm{Vol}^{d-1}(x_0)$$
$$= \int_{\mathcal{S}} \int_{-\epsilon_{n,s}^M}^{\epsilon_{n,s}^M} t \|\dot{\psi}(x_0)\| \Big\{ \Phi\big[2\sqrt{s}\big(\mathbb{P}\big(S_n(x_0^t) < 1/2\big) - 1/2\big)\big]$$
$$- \mathbb{1}\{t < 0\} \Big\} dt d\mathrm{Vol}^{d-1}(x_0) + R_{12}.$$

If $|\mathbb{P}\big(S_n(x_0^t) < 1/2\big) - 1/2| \leq \log(s)/\sqrt{s}$, along with Lemma 2, we have

$$\Big| \Phi\Big[\frac{\sqrt{s}\big(\mathbb{P}\big(S_n(x_0^t) < 1/2\big) - 1/2\big)}{\sqrt{\mathbb{P}\big(S_n(x_0^t) < 1/2\big)\mathbb{P}\big(S_n(x) \geq 1/2\big)}}\Big]$$
$$- \Phi\big[2\sqrt{s}\big(\mathbb{P}\big(S_n(x_0^t) < 1/2\big) - 1/2\big)\big] \Big|$$
$$\leq \sqrt{s}\big|\mathbb{P}\big(S_n(x_0^t) < 1/2\big) - 1/2\big|\Big|\frac{1}{\sqrt{1/4 + O(\frac{\log(s)}{\sqrt{s}})}} - 2\Big|$$
$$\leq \sqrt{s}\frac{\log(s)}{\sqrt{s}}\frac{1 - 2\sqrt{1/4 + O(\frac{\log(s)}{\sqrt{s}})}}{\sqrt{1/4 + O(\frac{\log(s)}{\sqrt{s}})}} = O(\frac{(\log(s))^2}{\sqrt{s}}).$$

In addition, if $|\mathbb{P}\big(S_n(x_0^t) \geq 1/2\big) - 1/2| > \frac{\log(s)}{\sqrt{s}}$, applying Lemma 3, we have

$$\Big|1 - \Phi\Big[\frac{\sqrt{s}\big|\mathbb{P}\big(S_n(x_0^t) < 1/2\big) - 1/2\big|}{\sqrt{\mathbb{P}\big(S_n(x_0^t) < 1/2\big)\mathbb{P}\big(S_n(x_0^t) \geq 1/2\big)}}\Big]\Big|$$
$$\leq \Big|1 - \Phi\big(2\log(s)\big)\Big| \leq \frac{1}{2\log(s)}\frac{e^{-[2\log(s)]^2/2}}{\sqrt{2\pi}} = o\Big(\frac{1}{\sqrt{s}}\Big),$$
$$\Big|1 - \Phi\big[2\sqrt{s}\big|\mathbb{P}\big(S_n(x_0^t) < 1/2\big) - 1/2\big|\big]\Big|$$
$$\leq \Big|1 - \Phi\big(2\log(s)\big)\Big| \leq \frac{1}{2\log(s)}\frac{e^{-[2\log(s)]^2/2}}{\sqrt{2\pi}} = o\Big(\frac{1}{\sqrt{s}}\Big).$$

In this case, we have

$$\Big|\Phi\Big[\frac{\sqrt{s}\big(\mathbb{P}\big(S_n(x_0^t) < 1/2\big) - 1/2\big)}{\sqrt{\mathbb{P}\big(S_n(x_0^t) < 1/2\big)\mathbb{P}\big(S_n(x) \geq 1/2\big)}}\Big]$$
$$- \Phi\big[2\sqrt{s}\big(\mathbb{P}\big(S_n(x_0^t) < 1/2\big) - 1/2\big)\big]\Big|$$
$$\leq \Big|1 - \Phi\Big[\frac{\sqrt{s}\big|\mathbb{P}\big(S_n(x_0^t) < 1/2\big) - 1/2\big|}{\sqrt{\mathbb{P}\big(S_n(x_0^t) < 1/2\big)\mathbb{P}\big(S_n(x_0^t) \geq 1/2\big)}}\Big]\Big|$$
$$+ \Big|1 - \Phi\big[2\sqrt{s}\big|\mathbb{P}\big(S_n(x_0^t) < 1/2\big) - 1/2\big|\big]\Big| = o(1/\sqrt{s}).$$

In summary, we have

$$\sup_{x_0 \in \mathcal{S}} \sup_{t \in [-\epsilon_{n,s}^M, \epsilon_{n,s}^M]} \Big|\Phi\Big[\frac{\sqrt{s}\big(\mathbb{P}\big(S_n(x_0^t) < 1/2\big) - 1/2\big)}{\sqrt{\mathbb{P}\big(S_n(x_0^t) < 1/2\big)\mathbb{P}\big(S_n(x) \geq 1/2\big)}}\Big] \tag{S.13}$$
$$- \Phi\big[2\sqrt{s}\big(\mathbb{P}\big(S_n(x_0^t) < 1/2\big) - 1/2\big)\big]\Big| = O((\log(s))^2/\sqrt{s}).$$

Therefore,

$$
\begin{aligned}
|R_{12}| \leq & \int_{\mathcal{S}} \int_{-\epsilon_{n,s}^M}^{\epsilon_{n,s}^M} |t| \|\dot{\psi}(x_0)\| \Big| \Phi\Big[\frac{\sqrt{s}\big(1/2 - \mathbb{P}\big(S_n(x_0^t) \geq 1/2\big)\big)}{\sqrt{\mathbb{P}\big(S_n(x_0^t) < 1/2\big)\mathbb{P}\big(S_n(x_0^t) \geq 1/2\big)}}\Big] \\
& - \Phi\big[2\sqrt{s}\big(\mathbb{P}\big(S_n(x_0^t) < 1/2\big) - 1/2\big)\big]\Big| dt\, d\mathrm{Vol}^{d-1}(x_0) \\
= & \int_{\mathcal{S}} \int_{-\epsilon_{n,s}^M}^{\epsilon_{n,s}^M} |t| \|\dot{\psi}(x_0)\| \Big| \Phi\Big[\frac{\sqrt{s}\big(\mathbb{P}\big(S_n(x_0^t) < 1/2\big) - 1/2\big)}{\sqrt{\mathbb{P}\big(S_n(x_0^t) < 1/2\big)\mathbb{P}\big(S_n(x_0^t) \geq 1/2\big)}}\Big] \\
& - \Phi\big[2\sqrt{s}\big(\mathbb{P}\big(S_n(x_0^t) < 1/2\big) - 1/2\big)\big]\Big| dt\, d\mathrm{Vol}^{d-1}(x_0) \\
\leq & O\big(\frac{(\log(s))^2}{\sqrt{s}}\big) \int_{\mathcal{S}} \int_{-\epsilon_{n,s}^M}^{\epsilon_{n,s}^M} |t| \|\dot{\psi}(x_0)\| dt\, d\mathrm{Vol}^{d-1}(x_0) = o(t_n^2 + \frac{1}{s} s_n^2).
\end{aligned}
$$

Next, we decompose

$$
\begin{aligned}
& \int_{\mathcal{S}} \int_{-\epsilon_{n,s}^M}^{\epsilon_{n,s}^M} t\|\dot{\psi}(x_0)\| \big\{ \Phi\big[2\sqrt{s}\big(\mathbb{P}\big(S_n(x_0^t) < 1/2\big) - 1/2\big)\big] \quad\quad (\mathrm{S}.14) \\
& \hspace{4cm} - \mathbb{1}\{t < 0\} \big\} dt\, d\mathrm{Vol}^{d-1}(x_0) \\
= & \int_{\mathcal{S}} \int_{-\epsilon_{n,s}^M}^{\epsilon_{n,s}^M} t\|\dot{\psi}(x_0)\| \big\{ \Phi\big[2\sqrt{s}\big(\Phi\big(\frac{1/2 - \mu_n(x_0^t)}{\sigma_n(x_0^t)}\big) - 1/2\big)\big] \\
& \hspace{4cm} - \mathbb{1}\{t < 0\} \big\} dt\, d\mathrm{Vol}^{d-1}(x_0) + R_{13}.
\end{aligned}
$$

Applying Lemma 2 and (S.6), we have

$$
\begin{aligned}
& \sup_{x_0 \in \mathcal{S}} \sup_{t \in [-\epsilon_{n,s}^M, \epsilon_{n,s}^M]} \Big| \Phi\big[2\sqrt{s}\big(\mathbb{P}\big(S_n(x_0^t) < 1/2\big) - 1/2\big)\big] \quad\quad (\mathrm{S}.15) \\
& \hspace{4cm} - \Phi\big[2\sqrt{s}\big(\Phi\big(\frac{1/2 - \mu_n(x_0^t)}{\sigma_n(x_0^t)}\big) - 1/2\big)\big]\Big| \\
\leq & \sup_{x_0 \in \mathcal{S}} \sup_{t \in [-\epsilon_{n,s}^M, \epsilon_{n,s}^M]} \sqrt{s} \Big| \mathbb{P}\big(S_n(x_0^t) < 1/2\big) - \Phi\big(\frac{1/2 - \mu_n(x_0^t)}{\sigma_n(x_0^t)}\big)\Big| \\
\leq & o\Big(\sqrt{s}\frac{1}{\sqrt{s}(\log(s))^2}\Big) = o((\log(s))^{-2}).
\end{aligned}
$$

Hence,

$$
\begin{aligned}
|R_{13}| \leq & \int_{\mathcal{S}} \int_{-\epsilon_{n,s}^M}^{\epsilon_{n,s}^M} |t| \|\dot{\psi}(x_0)\| \Big| \Phi\Big[2\sqrt{s}\big(\mathbb{P}\big(S_n(x_0^t) < 1/2\big) - 1/2\big)\Big] \\
& \hspace{2cm} - \Phi\Big[2\sqrt{s}\big(\Phi\big(\frac{1/2 - \mu_n(x_0^t)}{\sigma_n(x_0^t)}\big) - 1/2\big)\big)\Big| dt\, d\mathrm{Vol}^{d-1}(x_0) \\
= & o((\log(s))^{-2}) \int_{\mathcal{S}} \int_{-\epsilon_{n,s}^M}^{\epsilon_{n,s}^M} |t| \|\dot{\psi}(x_0)\| dt\, d\mathrm{Vol}^{d-1}(x_0) = o(s_n^2/s + t_n^2).
\end{aligned}
$$

Next, we decompose

$$
\begin{aligned}
& \int_{\mathcal{S}} \int_{-\epsilon_{n,s}^M}^{\epsilon_{n,s}^M} t\|\dot{\psi}(x_0)\| \big\{ \Phi\big[2\sqrt{s}\big(\Phi\big(\frac{1/2 - \mu_n(x_0^t)}{\sigma_n(x_0^t)}\big) - 1/2\big)\big] \quad\quad (\mathrm{S}.16) \\
& \hspace{4cm} - \mathbb{1}\{t < 0\} \big\} dt\, d\mathrm{Vol}^{d-1}(x_0) \\
= & \int_{\mathcal{S}} \int_{-\epsilon_{n,s}^M}^{\epsilon_{n,s}^M} t\|\dot{\psi}(x_0)\| \big\{ \Phi\big(\frac{1/2 - \mu_n(x_0^t)}{\sqrt{\pi/(2s)}\sigma_n(x_0^t)}\big) \\
& \hspace{4cm} - \mathbb{1}\{t < 0\} \big\} dt\, d\mathrm{Vol}^{d-1}(x_0) + R_{14}.
\end{aligned}
$$

If $|\frac{1/2-\mu_n(x_0^t)}{\sigma_n(x_0^t)}| \le \frac{\log(s)}{\sqrt{s}}$, applying Lemma 1 and Lemma 2, we have, for large $s$,

$$\left| \Phi\Big[2\sqrt{s}\big(\Phi\big(\frac{1/2-\mu_n(x_0^t)}{\sigma_n(x_0^t)}\big)-1/2\big)\Big] - \Phi\Big(\frac{1/2-\mu_n(x_0^t)}{\sqrt{\pi/(2s)}\sigma_n(x_0^t)}\Big)\right|$$

$$\le \sqrt{s}\Big|\Phi\Big(\frac{1/2-\mu_n(x_0^t)}{\sigma_n(x_0^t)}\Big)-1/2-\frac{1}{\sqrt{2\pi}}\Big(\frac{1/2-\mu_n(x_0^t)}{\sigma_n(x_0^t)}\Big)\Big|$$

$$= O\Big(\sqrt{s}\big(\frac{1/2-\mu_n(x_0^t)}{\sigma_n(x_0^t)}\big)^3\Big) = O\Big(\sqrt{s}\big(\frac{\log(s)}{\sqrt{s}}\big)^3\Big) = o\Big(\frac{1}{\sqrt{s}}\Big).$$

In addition, if $|\frac{1/2-\mu_n(x_0^t)}{\sigma_n(x_0^t)}| > \frac{\log(s)}{\sqrt{s}}$, applying mean value theorem, there exists $x_0 \in (0, \frac{\log(s)}{\sqrt{s}})$ such that, for large $s$

$$\Phi\Big(\Big|\frac{1/2-\mu_n(x_0^t)}{\sigma_n(x_0^t)}\Big|\Big) - 1/2 > \Phi\Big(\frac{\log(s)}{\sqrt{s}}\Big) - \Phi(0)$$

$$= \frac{\log(s)}{\sqrt{s}}\frac{1}{\sqrt{2\pi}}\exp(-x_0^2/2) > \frac{\log(s)}{\sqrt{s}}\frac{1}{\sqrt{2\pi}}\exp\Big(-(\frac{\log(s)}{\sqrt{s}})^2/2\Big) > \frac{\log(s)}{4\sqrt{s}}.$$

In this case, applying Lemma 3, we have for large $s$

$$1 - \Phi\Big(\Big|\frac{1/2-\mu_n(x_0^t)}{\sqrt{\pi/(2s)}\sigma_n(x_0^t)}\Big|\Big) < 1 - \Phi\big(\sqrt{2/\pi}\log(s)\big)$$

$$\le \frac{1}{\sqrt{2/\pi}\log(s)}\frac{e^{-[\sqrt{2/\pi}\log(s)]^2/2}}{\sqrt{2\pi}} = o\Big(\frac{1}{\sqrt{s}}\Big) \quad \text{and}$$

$$1 - \Phi\Big[2\sqrt{s}\Big(\Phi\Big(\Big|\frac{1/2-\mu_n(x_0^t)}{\sigma_n(x_0^t)}\Big|\Big)-1/2\Big)\Big] < 1 - \Phi\big((1/2)\log(s)\big)$$

$$\le \frac{1}{(1/2)\log(s)}\frac{e^{-[(1/2)\log(s)]^2/2}}{\sqrt{2\pi}} = o\Big(\frac{1}{\sqrt{s}}\Big).$$

Therefore,

$$\left| \Phi\Big[2\sqrt{s}\big(\Phi\big(\frac{1/2-\mu_n(x_0^t)}{\sigma_n(x_0^t)}\big)-1/2\big)\Big] - \Phi\Big(\frac{1/2-\mu_n(x_0^t)}{\sqrt{\pi/(2s)}\sigma_n(x_0^t)}\Big)\right|$$

$$= \left| \Phi\Big[2\sqrt{s}\Big(\Phi\Big(\Big|\frac{1/2-\mu_n(x_0^t)}{\sigma_n(x_0^t)}\Big|\Big)-1/2\Big)\Big] - \Phi\Big(\Big|\frac{1/2-\mu_n(x_0^t)}{\sqrt{\pi/(2s)}\sigma_n(x_0^t)}\Big|\Big)\right|$$

$$\le \left| 1 - \Phi\Big[2\sqrt{s}\Big(\Phi\Big(\Big|\frac{1/2-\mu_n(x_0^t)}{\sigma_n(x_0^t)}\Big|\Big)-1/2\Big)\Big]\right|$$

$$+ \left| 1 - \Phi\Big(\Big|\frac{1/2-\mu_n(x_0^t)}{\sqrt{\pi/(2s)}\sigma_n(x_0^t)}\Big|\Big)\right| = o\Big(\frac{1}{\sqrt{s}}\Big).$$

In summary, we have,

$$\sup_{x_0 \in \mathcal{S}} \sup_{t \in [-\epsilon_{n,s}^M, \epsilon_{n,s}^M]} \left| \Phi\Big[2\sqrt{s}\big(\Phi\big(\frac{1/2-\mu_n(x_0^t)}{\sigma_n(x_0^t)}\big)-1/2\big)\Big] \right. \tag{S.17}$$

$$\left. - \Phi\Big(\frac{1/2-\mu_n(x_0^t)}{\sqrt{\pi/(2s)}\sigma_n(x_0^t)}\Big)\right| = o\Big(\frac{1}{\sqrt{s}}\Big).$$

Therefore,

$$|R_{14}| \le \int_{\mathcal{S}}\int_{-\epsilon_{n,s}^M}^{\epsilon_{n,s}^M} |t|\|\dot\psi(x_0)\|\Big|\Phi\Big[2\sqrt{s}\big(\Phi\big(\frac{1/2-\mu_n(x_0^t)}{\sigma_n(x_0^t)}\big)-1/2\big)\Big]$$

$$- \Phi\Big(\frac{1/2-\mu_n(x_0^t)}{\sqrt{\pi/(2s)}\sigma_n(x_0^t)}\Big)\Big|dtd\mathrm{Vol}^{d-1}(x_0)$$

$$\le o\Big(\frac{1}{\sqrt{s}}\Big)\int_{\mathcal{S}}\int_{-\epsilon_{n,s}^M}^{\epsilon_{n,s}^M}|t|\|\dot\psi(x_0)\|dtd\mathrm{Vol}^{d-1}(x_0) = o(t_n^2 + \frac{1}{s}s_n^2).$$

Next, we decompose

$$\int_{\mathcal{S}}\int_{-\epsilon_{n,s}^M}^{\epsilon_{n,s}^M} t\|\dot{\psi}(x_0)\|\big\{\Phi\Big(\frac{1/2-\mu_n(x_0^t)}{\sqrt{\pi/(2s)}\sigma_n(x_0^t)}\Big)-\mathbb{1}\{t<0\}\big\}dtd\mathrm{Vol}^{d-1}(x_0) \quad (S.18)$$

$$= \int_{\mathcal{S}}\int_{-\epsilon_{n,s}^M}^{\epsilon_{n,s}^M} t\|\dot{\psi}(x_0)\|\big\{\Phi\Big(\frac{-2t\|\dot{\eta}(x_0)\|-2a(x_0)t_n}{\sqrt{\pi/(2s)}s_n}\Big)$$

$$-\mathbb{1}\{t<0\}\big\}dtd\mathrm{Vol}^{d-1}(x_0)+R_{15}.$$

Denote $r=t/s_n$ and $r_{x_0}=\frac{-a(x_0)t_n}{\|\dot{\eta}(x_0)s_n\|}$. According to $(S.1)$ and $(S.2)$, for a sufficiently small $\epsilon\in(0,\inf_{x_0\in\mathcal{S}}\|\dot{\eta}(x_0)\|)$ and a large $n$, for all $\boldsymbol{w}_n\in W_{n,\beta}$, $x_0\in\mathcal{S}$ and $r\in[-\epsilon_n/s_n,\epsilon_n/s_n]$, Samworth [5] showed that

$$\Big|\frac{1/2-\mu_n(x_0^{rs_n})}{\sigma_n(x_0^{rs_n})}-[-2\|\dot{\eta}(x_0)\|(r-r_{x_0})]\Big|\le\epsilon^2(|r|+t_n/s_n).$$

To adapt this to our setting, we need to scale some terms properly. Let $r^M=r\sqrt{2s/\pi}$, $s_{n,s}^M=s_n\sqrt{\pi/(2s)}$ and $r_{x_0}^M=r_{x_0}\sqrt{2s/\pi}=\sqrt{\frac{2s}{\pi}}\frac{-a(x_0)t_n}{\|\dot{\eta}(x_0)s_n\|}$, we have, when $r^M\in[-\epsilon_n/s_{n,s}^M,\epsilon_n/s_{n,s}^M]$,

$$\Big|\frac{1/2-\mu_n(x_0^{r^Ms_{n,s}^M})}{\sqrt{\pi/(2s)}\sigma_n(x_0^{r^Ms_{n,s}^M})}-[-2\|\dot{\eta}(x_0)\|(r^M-r_{x_0}^M)]\Big|$$

$$= \sqrt{2s/\pi}\Big|\frac{1/2-\mu_n(x_0^{rs_n})}{\sigma_n(x_0^{rs_n})}-[-2\|\dot{\eta}(x_0)\|(r-r_{x_0})]\Big|$$

$$\le \sqrt{2s/\pi}\epsilon^2(|r|+t_n/s_n)=\epsilon^2(|r^M|+t_n/s_{n,s}^M).$$

In addition, when $|r^M|\le\epsilon t_n/s_{n,s}^M$,

$$\Big|\Phi\Big(\frac{1/2-\mu_n(x_0^{r^Ms_{n,s}^M})}{\sqrt{\pi/(2s)}\sigma_n(x_0^{r^Ms_{n,s}^M})}\Big)-\Phi\big(-2\|\dot{\eta}(x_0)\|(r^M-r_{x_0}^M)\big)\Big|\le 1$$

and when $\epsilon t_n/s_{n,s}^M<|r^M|<\epsilon_n/s_{n,s}^M$,

$$\Big|\Phi\Big(\frac{1/2-\mu_n(x_0^{r^Ms_{n,s}^M})}{\sqrt{\pi/(2s)}\sigma_n(x_0^{r^Ms_{n,s}^M})}\Big)-\Phi\big(-2\|\dot{\eta}(x_0)\|(r^M-r_{x_0}^M)\big)\Big|$$

$$\le \epsilon^2(|r^M|+t_n/s_{n,s}^M)\phi(\|\dot{\eta}(x_0)\||r^M-r_{x_0}^M|),$$

where $\phi$ is the density function of standard normal distribution.

Therefore, after substituting $t=r^Ms_{n,s}^M$, we have

$$\int_{-\epsilon_{n,s}^M}^{\epsilon_{n,s}^M}|t|\|\dot{\psi}(x_0)\|\Big|\Phi\Big(\frac{1/2-\mu_n(x_0^t)}{\sqrt{\pi/(2s)}\sigma_n(x_0^t)}\Big)-\Phi\Big(\frac{-2t\|\dot{\eta}(x_0)\|-2a(x_0)t_n}{\sqrt{\pi/(2s)}s_n}\Big)\Big|dt$$

$$=\|\dot{\psi}(x_0)\|(s_{n,s}^M)^2\int_{-\epsilon_{n,s}^M/s_{n,s}^M}^{\epsilon_{n,s}^M/s_{n,s}^M}|r^M|\Big|\Phi\Big(\frac{1/2-\mu_n(x_0^{r^Ms_{n,s}^M})}{\sqrt{\pi/(2s)}\sigma_n(x_0^{r^Ms_{n,s}^M})}\Big)$$

$$-\Phi\big(-2\|\dot{\eta}(x_0)\|(r^M-r_{x_0}^M)\big)\Big|dr^M$$

$$\le\|\dot{\psi}(x_0)\|(s_{n,s}^M)^2\Big[\int_{|r^M|\le\epsilon t_n/s_{n,s}^M}|r^M|dr^M$$

$$+\epsilon^2\int_{-\infty}^{\infty}|r^M|(|r^M|+t_n/s_{n,s}^M)\phi(\|\dot{\eta}(x_0)\||r^M-r_{x_0}^M|)dr^M\Big]=o\Big(\frac{s_n^2}{s}+t_n^2\Big).$$

The inequality above leads to $R_{15}=o(s_n^2/s+t_n^2)$.

Combining (S.9), (S.10), (S.12), (S.14), (S.16) and (S.18), we have

$$\int_{\mathcal{S}} \int_{-\epsilon_{n,s}^M}^{\epsilon_{n,s}^M} \psi(x_0^t) \big\{ \mathbb{P}\big(S_{n,s}^M(x_0^t) < 1/2\big) - \mathbb{1}\{t < 0\} \big\} dt d\mathrm{Vol}^{d-1}(x_0) \qquad \text{(S.19)}$$

$$= \int_{\mathcal{S}} \int_{-\epsilon_{n,s}^M}^{\epsilon_{n,s}^M} t \|\dot\psi(x_0)\| \big\{ \Phi\Big( \frac{-2t\|\dot\eta(x_0)\| - 2a(x_0)t_n}{\sqrt{\pi/(2s)}s_n} \Big)$$

$$- \mathbb{1}\{t < 0\} \big\} dt d\mathrm{Vol}^{d-1}(x_0) + o\big(\frac{s_n^2}{s} + t_n^2\big).$$

Finally, after substituting $t = \sqrt{\pi/(2s)}us_n/2$ in (S.19), we have, up to $o(s_n^2/s + t_n^2)$ difference,

$$\mathrm{Regret}(\widehat\phi_{n,s}^M) = \frac{\pi}{8s} s_n^2 \int_{\mathcal{S}} \int_{-\infty}^{\infty} u \|\dot\psi(x_0)\| \big\{ \Phi\big( -\|\dot\eta(x_0)\|u - \frac{2a(x_0)t_n}{\sqrt{\pi/(2s)}s_n} \big)$$

$$- \mathbb{1}\{u < 0\} \big\} du d\mathrm{Vol}^{d-1}(x_0)$$

$$= \frac{\pi}{4s} s_n^2 \int_{\mathcal{S}} \int_{-\infty}^{\infty} u \|\dot\eta(x_0)\| \bar f(x_0) \big\{ \Phi\big( -\|\dot\eta(x_0)\|u - \frac{2a(x_0)t_n}{\sqrt{\pi/(2s)}s_n} \big) \qquad \text{(S.20)}$$

$$- \mathbb{1}\{u < 0\} \big\} du d\mathrm{Vol}^{d-1}(x_0)$$

$$= B_1 \frac{\pi}{2s} s_n^2 + B_2 t_n^2. \qquad \text{(S.21)}$$

(S.20) holds by Lemma 4, and (S.21) can be calculated by applying Lemma 5. This concludes the proof of Theorem 1. ∎

## 5  Proof of Theorem 3

In this section, we apply similar notations as those in Section 4. For the sake of simplicity, we omit $\boldsymbol{w}_n$ in the subscript of such notations as $\widehat\phi_{n,s,\boldsymbol{w}_n}^W$ and $S_{n,s,\boldsymbol{w}_n}^W$. We have

$$\mathrm{Regret}(\widehat\phi_{n,s}^W) = \int_{\mathcal{R}} \big[ \mathbb{P}\big(\widehat\phi_{n,s}^W(x) = 0\big) - \mathbb{1}\{\eta(x) < 1/2\} \big] dP^\circ(x).$$

Denote the average of estimated regression function from $s$ subsamples as

$$S_{n,s}^W(x) = s^{-1} \sum_{j=1}^s S_n^{(j)}(x).$$

We can also write $S_{n,s}^W(x)$ as

$$S_{n,s}^W(x) = s^{-1} \sum_{j=1}^s \sum_{i=1}^n w_{ni} Y_{(i)}^{(j)} = \sum_{l=1}^N w_{Nl} Y_l,$$

where

$$\{Y_1, Y_2, \dots Y_N\} = \{Y_{(1)}^{(1)}, Y_{(1)}^{(2)}, \dots, Y_{(1)}^{(s)}, \dots, Y_{(n)}^{(1)}, Y_{(n)}^{(2)}, \dots, Y_{(n)}^{(s)}\},$$

$$\{w_{N1}, w_{N2}, \dots w_{NN}\} = \{ \frac{w_{n1}}{s}, \frac{w_{n1}}{s} \dots, \frac{w_{n1}}{s}, \dots, \frac{w_{nn}}{s}, \frac{w_{nn}}{s}, \dots, \frac{w_{nn}}{s} \}.$$

The W-DiNN classifier is defined as

$$\widehat\phi_{n,s}^W(x) = \mathbb{1}\big\{ S_{n,s}^W(x) \geq 1/2 \big\}.$$

Since $\mathbb{P}\big(\widehat\phi_{n,s}^W(x) = 0\big) = \mathbb{P}\big(S_{n,s}^W(x) < 1/2\big)$, the regret of W-DiNN becomes

$$\mathrm{Regret}(\widehat\phi_{n,s}^W) = \int_{\mathcal{R}} \big\{ \mathbb{P}(S_{n,s}^W(x) < 1/2) - \mathbb{1}\{\eta(x) < 1/2\} \big\} dP^\circ(x).$$

Let $\mu_{n,s}(x) = \mathbb{E}\{S_{n,s}^W(x)\}$, $\sigma_{n,s}^2(x) = \mathrm{Var}\{S_{n,s}^W(x)\}$. We have

$$\mu_{n,s}(x) = \mathbb{E}\{S_{n,s}^W(x)\} = \mathbb{E}\{s^{-1} \sum_{j=1}^s S_n^{(j)}(x)\} = \mu_n(x),$$

$$\sigma_{n,s}^2(x) = \mathrm{Var}\{S_{n,s}^W(x)\} = \mathrm{Var}\{s^{-1} \sum_{j=1}^s S_n^{(j)}(x)\} = s^{-1}\sigma_n^2(x).$$

Denote $\epsilon_{n,s}^W = \epsilon_n/\sqrt{s}$, $s_{n,s}^2 = s_n^2/s$ and $t_{n,s} = t_n$. We have, uniformly for $\boldsymbol{w}_n \in W_{n,\beta}$,

$$\sup_{x \in \mathcal{S}^{\epsilon_n}} |\mu_{n,s}(x) - \eta(x) - a(x)t_{n,s}| = \sup_{x \in \mathcal{S}^{\epsilon_n}} |\mu_n(x) - \eta(x) - a(x)t_n|$$

$$= o(t_n) = o(t_{n,s}),$$

$$\sup_{x \in \mathcal{S}^{\epsilon_n}} \left|\sigma_{n,s}^2(x) - \frac{1}{4}s_{n,s}^2\right| = \sup_{x \in \mathcal{S}^{\epsilon_n}} \left|\frac{\sigma_n^2(x)}{s} - \frac{1}{4}\frac{s_n^2}{s}\right|$$

$$= o(s_n^2/s) = o(s_{n,s}^2).$$

We organize our proof in three steps. In *Step 1*, we decompose the integral over $\mathcal{R} \cap \mathcal{S}^{\epsilon_n}$ as an integral along $\mathcal{S}$ and an integral in the perpendicular direction; in *Step 2*, we focus on the complement set $\mathcal{R}\backslash\mathcal{S}^{\epsilon_n}$; *Step 3* combines the results and applies a normal approximation in $\mathcal{S}^{\epsilon_n}$ to yield the final conclusion.

*Step 1*: Similarly to *Step 1* in Section 4, we have

$$\int_{\mathcal{R}\cap\mathcal{S}^{\epsilon_n}} \left\{\mathbb{P}(S_{n,s}^W(x) < 1/2) - \mathbb{1}\{\eta(x) < 1/2\}\right\}dP^\circ(x)$$

$$= \int_{\mathcal{S}} \int_{-\epsilon_n}^{\epsilon_n} \psi(x_0^t)\left\{\mathbb{P}(S_{n,s}^W(x_0^t) < 1/2) - \mathbb{1}\{t < 0\}\right\}dtd\mathrm{Vol}^{d-1}(x_0)\{1 + o(1)\},$$

uniformly for $\boldsymbol{w}_n \in W_{n,\beta}$.

*Step 2*: Bound the contribution to regret from $\mathcal{R}\backslash\mathcal{S}^{\epsilon_n}$. We show that

$$\sup_{\boldsymbol{w}_n \in W_{n,\beta}} \int_{\mathcal{R}\backslash\mathcal{S}^{\epsilon_n}} \left\{\mathbb{P}(S_{n,s}^W(x) < 1/2) - \mathbb{1}\{\eta(x) < 1/2\}\right\}dP^\circ(x) = o\left(\frac{s_n^2}{s} + t_n^2\right).$$

Samworth [5] showed that, in any subsample, there exists a constant $c_{30} > 0$ such that, for a sufficiently large $n$,

$$\inf_{x \in \mathcal{R}\backslash\mathcal{S}^{\epsilon_n}} |\mu_n(x) - 1/2| \geq c_{30}\epsilon_n/4.$$

Applying Hoeffding's inequality to $S_{n,s}^W(x)$, we have

$$|\mathbb{P}(S_{n,s}^W(x) < 1/2) - \mathbb{1}\{\eta(x) < 1/2\}| \leq \exp\left(\frac{-2(\mu_{n,s}(x) - 1/2)^2}{\sum_{l=1}^N (w_{Nl} - 0)^2}\right)$$

$$= \exp\left(\frac{-2(\mu_n(x) - 1/2)^2}{s_n^2/s}\right) \leq \exp\left(\frac{-2s(c_{30}\epsilon_n/4)^2}{n^{-\beta}}\right) = o\left(\frac{s_n^2}{s} + t_n^2\right),$$

uniformly for $\boldsymbol{w}_n \in W_{n,\beta}$ and $x \in \mathcal{R}\backslash\mathcal{S}^{\epsilon_n}$.

*Step 3*: In the end, we will show

$$\int_{\mathcal{S}} \int_{-\epsilon_n}^{\epsilon_n} \psi(x_0^t)\left\{\mathbb{P}(S_{n,s}^W(x_0^t) < 1/2) - \mathbb{1}\{t < 0\}\right\}dtd\mathrm{Vol}^{d-1}(x_0)$$

$$= B_1\frac{s_n^2}{s} + B_2 t_n^2 + o\left(\frac{s_n^2}{s} + t_n^2\right).$$

According to (S.8), we have

$$\int_{\mathcal{S}} \int_{-\epsilon_n}^{\epsilon_n} \psi(x_0^t)\left\{\mathbb{P}(S_{n,s}^W(x_0^t) < 1/2) - \mathbb{1}\{t < 0\}\right\}dtd\mathrm{Vol}^{d-1}(x_0) \quad\quad\text{(S.22)}$$

$$= \int_{\mathcal{S}} \int_{-\epsilon_n}^{\epsilon_n} t\|\dot{\psi}(x_0)\|\left\{\mathbb{P}(S_{n,s}^W(x_0^t) < 1/2) \right.$$

$$\left. -\mathbb{1}\{t < 0\}\right\}dtd\mathrm{Vol}^{d-1}(x_0)\{1 + o(1)\}.$$

Next, we decompose

$$\int_{\mathcal{S}}\int_{-\epsilon_n}^{\epsilon_n} t\|\dot\psi(x_0)\|\big\{\mathbb{P}\big(S_{n,s}^W(x_0^t)<1/2\big)-\mathbb{1}\{t<0\}\big\}dtd\text{Vol}^{d-1}(x_0) \qquad (\text{S.23})$$

$$=\int_{\mathcal{S}}\int_{-\epsilon_n}^{\epsilon_n} t\|\dot\psi(x_0)\|\Big\{\Phi\big(\frac{1/2-\mu_{n,s}(x_0^t)}{\sigma_{n,s}(x_0^t)}\big)$$
$$-\mathbb{1}\{t<0\}\big\}dtd\text{Vol}^{d-1}(x_0)+R_{31}.$$

Let $Z_l=(w_{Nl}Y_l-w_{Nl}\mathbb{E}[Y_l])/\sigma_{n,s}(x)$ and $V=\sum_{l=1}^N Z_l$. Note that $\mathbb{E}(Z_l)=0$, $\text{Var}(Z_l)<\infty$, and $\text{Var}(V)=1$. The nonuniform Berry-Esseen Theorem [3] implies that there exists a constant $c_{31}>0$, such that

$$\Big|\mathbb{P}(V\le By)-\Phi(y)\Big|\le\frac{c_{31}A}{B^3(1+|y|^3)},$$

where $A=\sum_{l=1}^N E|Z_l|^3$ and $\big(\sum_{l=1}^N E|Z_l|^2\big)^{1/2}$. In the case of W-DiNN,

$$A=\sum_{l=1}^N\mathbb{E}\big|\frac{w_{Nl}Y_l-w_{Nl}\mathbb{E}[Y_l]}{\sigma_{n,s}^3(x)}\big|^3\le\sum_{l=1}^N\frac{16|w_{Nl}|^3}{s_{n,s}^3}=\frac{16\sum_{l=1}^N w_{Nl}^3}{s_{n,s}^3},$$
$$B=(\sum_{l=1}^N\text{Var}(Z_l))^{1/2}=\sqrt{\text{Var}(V)}=1.$$

Denote $c_{32}=16c_{31}$, we have

$$\sup_{x_0\in\mathcal{S}}\sup_{t\in[-\epsilon_n,\epsilon_n]}\Big|\mathbb{P}\Big(\frac{S_{n,s}^W(x_0^t)-\mu_{n,s}(x_0^t)}{\sigma_{n,s}(x_0^t)}\le y\Big)-\Phi(y)\Big| \qquad (\text{S.24})$$

$$\le\frac{\sum_{l=1}^N w_{Nl}^3}{s_{n,s}^3}\frac{c_{32}}{1+|y|^3}.$$

[5] showed that, there exists constants $c_{33},c_{34}>0$ such that, uniformly for $\boldsymbol{w}_n\in W_{n,\beta}$,

$$\inf_{x_0\in\mathcal{S}}\inf_{c_{33}t_n\le|t|\le\epsilon_n}\Big|\frac{1/2-\mu_n(x_0^t)}{\sigma_n(x_0^t)}\Big|\ge\frac{c_{34}|t|}{s_n}.$$

Hence,

$$\inf_{x_0\in\mathcal{S}}\inf_{c_{33}t_n\le|t|\le\epsilon_n}\Big|\frac{1/2-\mu_{n,s}(x_0^t)}{\sigma_{n,s}(x_0^t)}\Big|\ge\frac{c_{34}|t|}{s_n/\sqrt{s}}=\frac{c_{34}|t|}{s_{n,s}}. \qquad (\text{S.25})$$

Therefore,

$$\int_{-\epsilon_n}^{\epsilon_n}|t|\|\dot\psi(x_0)\|\Big|\mathbb{P}\big(S_{n,s}^W(x_0^t)<1/2\big)-\Phi\Big(\frac{1/2-\mu_{n,s}(x_0^t)}{\sigma_{n,s}(x_0^t)}\Big)\Big|dt$$

$$\le\int_{|t|\le c_{33}t_n}|t|\|\dot\psi(x_0)\|\frac{c_{32}\sum_{l=1}^N w_{Nl}^3}{s_{n,s}^3}dt$$

$$+\int_{c_{33}t_n\le|t|\le\epsilon_n}\frac{c_{32}\sum_{l=1}^N w_{Nl}^3}{s_{n,s}^3}\frac{|t|\|\dot\psi(x_0)\|}{1+c_{34}^3|t|^3/s_{n,s}^3}dt$$

$$\le\frac{c_{32}\sum_{i=1}^n w_{ni}^3}{\sqrt{s}s_n^3}\int_{|t|\le c_{33}t_n}|t|\|\dot\psi(x_0)\|dt$$

$$+\frac{c_{32}\sum_{i=1}^n w_{ni}^3}{\sqrt{s}s_n^3}\int_{c_{33}t_n\le|t|\le\epsilon_n}\frac{\|\dot\psi(x_0)\|\,|t|}{c_{34}^2s|t|^2/s_n^2}dt=o(\frac{s_n^2}{s}+t_n^2).$$

The inequality above leads to $|R_{31}|=o(s_n^2/s+t_n^2)$.

Next, we decompose

$$\int_{\mathcal{S}}\int_{-\epsilon_n}^{\epsilon_n} t\|\dot\psi(x_0)\|\big\{\Phi\big(\frac{1/2-\mu_{n,s}(x_0^t)}{\sigma_{n,s}(x_0^t)}\big)-\mathbb{1}\{t<0\}\big\}dtd\text{Vol}^{d-1}(x_0) \qquad (\text{S.26})$$

$$=\int_{\mathcal{S}}\int_{-\epsilon_n}^{\epsilon_n} t\|\dot\psi(x_0)\|\big\{\Phi\big(\frac{-2t\|\dot\eta(x_0)\|-2a(x_0)t_n}{s_n/\sqrt{s}}\big)$$
$$-\mathbb{1}\{t<0\}\big\}dtd\text{Vol}^{d-1}(x_0)+R_{32}.$$

Denote $r^W = r\sqrt{s}$ and $r_{x_0}^W = r_{x_0}\sqrt{s}$. Similarly to bounding $R_{15}$ in (S.18), we have

$$\int_{-\epsilon_n}^{\epsilon_n} |t| \|\dot{\psi}(x_0)\| \Big| \Phi\Big(\frac{1/2 - \mu_{n,s}(x_0^t)}{\sigma_{n,s}(x_0^t)}\Big) - \Phi\Big(\frac{-2t\|\dot{\eta}(x_0)\| - 2a(x_0)t_n}{s_n/\sqrt{s}}\Big)\Big| dt$$

$$= \|\dot{\psi}(x_0)\| s_{n,s}^2 \int_{-\epsilon_n/s_{n,s}}^{\epsilon_n/s_{n,s}} |r^W| \Big| \Phi\Big(\frac{1/2 - \mu_n(x_0^{r^W s_{n,s}})}{s^{-1/2}\sigma_n(x_0^{r^W s_{n,s}})}\Big)$$

$$- \Phi\big(-2\|\dot{\eta}(x_0)\|(r^W - r_{x_0}^W)\big)\Big| dr^W$$

$$\leq \|\dot{\psi}(x_0)\| s_{n,s}^2 \Big[ \int_{|r^W| \leq \epsilon t_n/s_{n,s}} |r^W| dr^W$$

$$+ \epsilon^2 \int_{-\infty}^{\infty} |r^W|(|r^W| + t_n/s_{n,s})\phi(\|\dot{\eta}(x_0)\| |r^W - r_{x_0}^W|) dr^W \Big] = o\big(\frac{s_n^2}{s} + t_n^2\big).$$

The inequality above leads to $R_{32} = o(s_n^2/s + t_n^2)$.

By (S.22), (S.23) and (S.26), we have

$$\int_{\mathcal{S}} \int_{-\epsilon_n}^{\epsilon_n} \psi(x_0^t)\big\{ \mathbb{P}\big(S_{n,s}^W(x_0^t) < 1/2\big) - \mathbb{1}\{t < 0\}\big\} dt d\text{Vol}^{d-1}(x_0) \qquad \text{(S.27)}$$

$$= \int_{\mathcal{S}} \int_{-\epsilon_n}^{\epsilon_n} t\|\dot{\psi}(x_0)\|\big\{ \Phi\big(\frac{-2t\|\dot{\eta}(x_0)\| - 2a(x_0)t_n}{s_n/\sqrt{s}}\big)$$

$$- \mathbb{1}\{t < 0\}\big\} dt d\text{Vol}^{d-1}(x_0) + o(s_n^2/s + t_n^2).$$

Finally, after replacing $t = us_n/(2\sqrt{s})$ in (S.27), we have, up to $o(s_n^2/s + t_n^2)$ difference,

$$\text{Regret}(\widehat{\phi}_{n,s}^W) = \frac{s_n^2}{4s} \int_{\mathcal{S}} \int_{-\infty}^{\infty} \|\dot{\psi}(x_0)\| u\big\{ \Phi\big(-\|\dot{\eta}(x_0)\| u - \frac{2a(x_0)t_n}{s_n/\sqrt{s}}\big)$$

$$- \mathbb{1}\{u < 0\}\big\} du d\text{Vol}^{d-1}(x_0)$$

$$= \frac{s_n^2}{2s} \int_{\mathcal{S}} \int_{-\infty}^{\infty} \|\dot{\eta}(x_0)\| \bar{f}(x_0) u\big\{ \Phi\big(-\|\dot{\eta}(x_0)\| u - \frac{2a(x_0)t_n}{s_n/\sqrt{s}}\big) \qquad \text{(S.28)}$$

$$- \mathbb{1}\{u < 0\}\big\} du d\text{Vol}^{d-1}(x_0)$$

$$= B_1\frac{1}{s}s_n^2 + B_2 t_n^2. \qquad \text{(S.29)}$$

(S.28) holds by Lemma 4, and (S.29) can be calculated by Lemma 5. This completes the proof of Theorem 3. ∎

# 6 Proof of Theorem 2 and Theorem 4

From Theorem 1 and Proposition 1, we have, for large $n, s$,

$$\frac{\text{Regret}(\widehat{\phi}_{n,s,\boldsymbol{w}_n}^M)}{\text{Regret}(\widehat{\phi}_{N,\boldsymbol{w}_N})} = \frac{\big[B_1\frac{\pi}{2s}\sum_{i=1}^n w_{ni}^2 + B_2\big(\sum_{i=1}^n \frac{\alpha_i w_{ni}}{n^{2/d}}\big)^2\big]\{1 + o(1)\}}{\big[B_1\sum_{i=1}^N w_{Ni}^2 + B_2\big(\sum_{i=1}^N \frac{\alpha_i w_{Ni}}{N^{2/d}}\big)^2\big]\{1 + o(1)\}} \to \big(\frac{\pi}{2}\big)^{\frac{4}{d+4}}, \; as \; n, s \to \infty.$$

The last equality holds by (6) and (7). This completes the proof of Theorem 2.

Similarly, from Theorem 3 and Proposition 1, we have, for large $n, s$,

$$\frac{\text{Regret}(\widehat{\phi}_{n,s,\boldsymbol{w}_n}^W)}{\text{Regret}(\widehat{\phi}_{N,\boldsymbol{w}_N})} = \frac{\big[B_1\frac{1}{s}\sum_{i=1}^n w_{ni}^2 + B_2\big(\sum_{i=1}^n \frac{\alpha_i w_{ni}}{n^{2/d}}\big)^2\big]\{1 + o(1)\}}{\big[B_1\sum_{i=1}^N w_{Ni}^2 + B_2\big(\sum_{i=1}^N \frac{\alpha_i w_{Ni}}{N^{2/d}}\big)^2\big]\{1 + o(1)\}} \to 1, \; as \; n, s \to \infty.$$

The last equality holds by (12) and (13). This completes the proof of Theorem 4. ∎

# 7 Proof of Corollary 1

Denote $a_n \succeq b_n$ if $b_n = O(a_n)$, $a_n \succ b_n$ if $b_n = o(a_n)$, $a_n \asymp b_n$ if $a_n \succeq b_n$ and $b_n \succeq a_n$. To find the optimal value of (4), we write its Lagrangian as

$$L(\boldsymbol{w}_n) = \Big(\sum_{i=1}^{n} \frac{\alpha_i w_{ni}}{n^{2/d}}\Big)^2 + \lambda \sum_{i=1}^{n} w_{ni}^2 + \nu\Big(\sum_{i=1}^{n} w_{ni} - 1\Big),$$

where $\lambda = (\pi B_1)/(2sB_2)$. Since all the weights are nonnegative, we denote $l^* = \max\{i : w_{ni}^* > 0\}$. Setting the derivative of $L(\boldsymbol{w}_n)$ to be 0, we have

$$\frac{\partial L(\boldsymbol{w}_n)}{\partial w_{ni}} = 2n^{-4/d}\alpha_i \sum_{i=1}^{l^*} \alpha_i w_{ni} + 2\lambda w_{ni} + \nu = 0. \tag{S.30}$$

(i) Summing $(S.30)$ from 1 to $l^*$, (ii) multiplying $(S.30)$ by $\alpha_i$ and then summing from 1 to $l^*$, (iii) multiplying $(S.30)$ by $w_{ni}$ and then summing from 1 to $l^*$, we have

$$2n^{-4/d}(l^*)^{1+2/d}\sum_{i=1}^{l^*}\alpha_i w_{ni} + 2\lambda + \nu l^* = 0,$$
$$2n^{-4/d}\sum_{i=1}^{l^*}\alpha_i w_{ni}\sum_{i=1}^{l^*}\alpha_i^2 + 2\lambda\sum_{i=1}^{l^*}\alpha_i w_{ni} + \nu(l^*)^{1+2/d} = 0,$$
$$2n^{-4/d}\Big(\sum_{i=1}^{l^*}\alpha_i w_{ni}\Big)^2 + 2\lambda\sum_{i=1}^{l^*}w_{ni}^2 + \nu = 0.$$

Therefore, we have

$$w_{ni}^* = \frac{1}{l^*} + \frac{(l^*)^{4/d} - (l^*)^{2/d}\alpha_i}{\sum_{j=1}^{l^*}\alpha_j^2 + \lambda n^{4/d} - (l^*)^{1+4/d}}, \tag{S.31}$$

$$\sum_{i=1}^{l^*}\alpha_i w_{ni} \asymp (l^*)^{2/d}, \quad \text{and} \quad \sum_{i=1}^{l^*}w_{ni}^2 \asymp \frac{1}{l^*}. \tag{S.32}$$

Here $w_{ni}^*$ is decreasing in $i$, since $\alpha_i$ is increasing in $i$ and $\sum_{j=1}^{l^*}\alpha_j^2 + \lambda n^{4/d} - (l^*)^{1+4/d} > 0$ from Lemma 6. Next we solve for $l^*$. According to the definition of $l^*$, we only need to find the last $l$ such that $w_{nl}^* > 0$. Using the results from Lemma 6, solving this equation reduces to finding the $l^*$ such that

$$(1 + \frac{2}{d})(l^* - 1)^{2/d} \leq \lambda n^{4/d}(l^*)^{-1-2/d} + \frac{(d+2)^2}{d(d+4)}(l^*)^{2/d}\{1 + O(\frac{1}{l^*})\}$$
$$\leq (1 + \frac{2}{d})(l^*)^{2/d}.$$

For large $n, s$, we have

$$l^* = \Big\lceil \Big\{\frac{d(d+4)}{2(d+2)}\Big\}^{\frac{d}{d+4}} \lambda^{\frac{d}{d+4}} n^{\frac{4}{d+4}} \Big\rceil = \Big\lceil \Big\{\frac{d(d+4)}{2(d+2)}\Big\}^{\frac{d}{d+4}} \Big(\frac{\pi B_1}{2sB_2}\Big)^{\frac{d}{d+4}} n^{\frac{4}{d+4}} \Big\rceil.$$

Due to Assumption (w.1) in Section 2, we have $l^* \to \infty$ as $n \to \infty$. When $\gamma < 2/(d+4)$, plugging $l^*$ and $(S.37)$ into $(S.31)$ yields the optimal weight and (10).

Denote $H(\boldsymbol{w}_n)$ as the Hessian matrix of $L(\boldsymbol{w}_n)$. We have

$$\frac{\partial^2 L(\boldsymbol{w}_n)}{\partial w_{ni}^2} = 2n^{-4/d}\alpha_i^2 + 2\lambda \quad \text{and} \quad \frac{\partial^2 L(\boldsymbol{w}_n)}{\partial w_{ni}\partial w_{nj}} = 2n^{-4/d}\alpha_i\alpha_j.$$

For any nonzero vector $X_{l^*} = (x_1, ..., x_{l^*})^T$, we have

$$X_{l^*}^T H(\boldsymbol{w}_n)X_{l^*} = 2n^{-4/d}\sum_{i=1}^{l^*}\alpha_i^2 x_i^2 + 2\lambda\sum_{i=1}^{l^*}x_i^2 + 2n^{-4/d}\sum_{i\neq j}\alpha_i\alpha_j x_i x_j$$
$$= 2n^{-4/d}\Big(\sum_{i=1}^{l^*}\alpha_i x_i\Big)^2 + 2\lambda\sum_{i=1}^{l^*}x_i^2 > 0.$$

Therefore, $H(\boldsymbol{w}_n)$ is positive definite, and this verifies that the above optimal value achieves the global minimum.

Next, we analyze the case of $\gamma \geq 2/(d+4)$. By Cauchy–Schwarz inequality, we have

$$(\textstyle\sum_{i=1}^{l^*} \boldsymbol{w}_{ni}^3)(\sum_{i=1}^{l^*} \boldsymbol{w}_{ni}) \geq (\sum_{i=1}^{l^*} \boldsymbol{w}_{ni}^{3/2} \boldsymbol{w}_{ni}^{1/2})^2$$
$$= (\textstyle\sum_{i=1}^{l^*} \boldsymbol{w}_{ni}^2)^2 \geq (\sum_{i=1}^{l^*} \boldsymbol{w}_{ni}^2)^{3/2}/\sqrt{l^*}.$$

The above inequality, along with condition (3), suggests that $l^* \succ s$. As $\gamma \geq 2/(d+4)$, we have $l^* \succ s \succeq N^{2/(d+4)}$ and $n = O(N^{(d+2)/(d+4)})$. Applying (S.32), we have, as $n, s \to \infty$,

$$\Big( \sum_{i=1}^{n} \frac{\alpha_i w_{ni}}{n^{2/d}} \Big)^2 \asymp (l^*/n)^{4/d} \succ N^{-4/(d+4)}.$$

Samworth [5] showed that

$$\operatorname{Regret}(\widehat{\phi}_{N,\boldsymbol{w}_N^*}) \asymp N^{-4/(d+4)}. \tag{S.33}$$

Therefore, we have, as $n, s \to \infty$,

$$\frac{\operatorname{Regret}(\widehat{\phi}_{n,s,w_n}^M)}{\operatorname{Regret}(\widehat{\phi}_{N,\boldsymbol{w}_N^*})} \asymp \frac{B_1 \frac{\pi}{2s} \sum_{i=1}^{n} w_{ni}^2 + B_2 \big( \sum_{i=1}^{n} \frac{\alpha_i w_{ni}}{n^{2/d}} \big)^2}{N^{-4/(d+4)}}$$

$$\succeq \frac{B_2 \big( \sum_{i=1}^{n} \frac{\alpha_i w_{ni}}{n^{2/d}} \big)^2}{N^{-4/(d+4)}} \to \infty.$$

This completes the proof of Corollary 1. ∎

# 8 Proof of Corollary 2

To find the optimal value of (11), we write its Lagrangian as

$$L(\boldsymbol{w}_n) = \Big( \sum_{i=1}^{n} \frac{\alpha_i w_{ni}}{n^{2/d}} \Big)^2 + \delta \sum_{i=1}^{n} w_{ni}^2 + \nu(\sum_{i=1}^{n} w_{ni} - 1),$$

where $\delta = (B_1)/(sB_2)$.

Similar to Section 7, replacing $l^*$ by $l^\dagger$ in the optimization, we have

$$w_{ni}^\dagger = \frac{1}{l^\dagger} + \frac{(l^\dagger)^{4/d} - (l^\dagger)^{2/d} \alpha_i}{\sum_{j=1}^{l^\dagger} \alpha_j^2 + \delta n^{4/d} - (l^\dagger)^{1+4/d}}. \tag{S.34}$$

For large $n, s$, we have

$$l^\dagger = \Big\lceil \Big\{ \frac{d(d+4)}{2(d+2)} \Big\}^{\frac{d}{d+4}} \delta^{\frac{d}{d+4}} n^{\frac{4}{d+4}} \Big\rceil = \Big\lceil \Big\{ \frac{d(d+4)}{2(d+2)} \Big\}^{\frac{d}{d+4}} \Big( \frac{B_1}{sB_2} \Big)^{\frac{d}{d+4}} n^{\frac{4}{d+4}} \Big\rceil.$$

Due to Assumption (w.1) in Section 2, we have $l^\dagger \to \infty$ as $n \to \infty$. When $\gamma < 4/(d+4)$, plugging $l^\dagger$ and (S.37) into (S.34) yields the optimal weight and (15).

When $\gamma \geq 4/(d+4)$, we have $s \succeq N^{4/(d+4)}$ and $n = O(N^{d/(d+4)})$. Similary to (S.32), we have, as $n, s \to \infty$,

$$\Big( \sum_{i=1}^{n} \frac{\alpha_i w_{ni}}{n^{2/d}} \Big)^2 \asymp \Big( \frac{l^\dagger}{n} \Big)^{4/d} \succeq (l^\dagger)^{4/d} N^{-4/(d+4)} \succ N^{-4/(d+4)}.$$

The last inequality holds by $l^\dagger \to \infty$ as $n \to \infty$. Therefore, along with (S.33), we have, as $n, s \to \infty$,

$$\frac{\operatorname{Regret}(\widehat{\phi}_{n,s,w_n}^W)}{\operatorname{Regret}(\widehat{\phi}_{N,\boldsymbol{w}_N^*})} \asymp \frac{B_1 \frac{\pi}{2s} \sum_{i=1}^{n} w_{ni}^2 + B_2 \big( \sum_{i=1}^{n} \frac{\alpha_i w_{ni}}{n^{2/d}} \big)^2}{N^{-4/(d+4)}}$$

$$\succeq \frac{B_2 \big( \sum_{i=1}^{n} \frac{\alpha_i w_{ni}}{n^{2/d}} \big)^2}{N^{-4/(d+4)}} \to \infty.$$

This completes the proof of Corollary 2. ∎

## 9  Lemmas

In this section, we provide some lemmas.

- Lemma 1–Lemma 5 are used for proving Theorem 1.
- Lemma 6 is used for proving Corollary 1.

**Lemma 1** *When $x$ is close to $0$ enough, we have*

$$\Phi(x) - 1/2 = \frac{1}{\sqrt{2\pi}}x + O(x^3),$$

*where $\Phi(x)$ is the standard normal distribution function.*

Proof of Lemma 1: When $x$ is close to $0$ enough, by Taylor expansion of $\Phi(x)$ at $0$, we have

$$
\begin{aligned}
\Phi(x) &= \Phi(0) + \Phi'(0)x + \frac{1}{2}\Phi''(0)x^2 + O(\Phi'''(0)x^3) \\
&= \frac{1}{2} + \frac{1}{\sqrt{2\pi}}x + O(x^3). \blacksquare
\end{aligned}
$$

**Lemma 2** *For constant $a > 0$, we have*

$$|\Phi(ax_1) - \Phi(ax_2)| \leq (a/2)|x_1 - x_2|, \tag{S.35}$$

*where $\Phi(x)$ is the standard normal distribution function.*

Proof of Lemma 2: If $x_1 = x_2$, (S.35) holds obviously.

If $x_1 < x_2$, by mean value theorem, there exists $x_0 \in (x_1, x_2)$ such that

$$\Phi(ax_1) - \Phi(ax_2) = \frac{1}{\sqrt{2\pi}}e^{-(ax_0)^2/2}a(x_1 - x_2).$$

Therefore,

$$|\Phi(ax_1) - \Phi(ax_2)| = \frac{1}{\sqrt{2\pi}}\exp\Big(-\frac{(ax_0)^2}{2}\Big)a|x_1 - x_2| \leq (a/2)|x_1 - x_2|.$$

Similarly, we can derive (S.35) when $x_1 > x_2$. $\blacksquare$

**Lemma 3** *[1] For all $x > 0$, we have*

$$1 - \Phi(x) = \int_x^\infty \frac{1}{\sqrt{2\pi}}e^{-t^2/2}dt \leq \frac{1}{x}\frac{e^{-x^2/2}}{\sqrt{2\pi}}.$$

Proof of Lemma 3:

$$\int_x^\infty \frac{1}{\sqrt{2\pi}}e^{-t^2/2}dt \leq \int_x^\infty \frac{t}{x}\frac{1}{\sqrt{2\pi}}e^{-t^2/2}dt = \frac{1}{x}\frac{e^{-x^2/2}}{\sqrt{2\pi}}. \blacksquare$$

**Lemma 4** *For $x_0 \in \mathcal{S}$, we have*

$$2\bar{f}(x_0)\|\dot{\eta}(x_0)\| = \|\dot{\psi}(x_0)\| \text{ and } \dot{\psi}(x_0)^T\dot{\eta}(x_0) = \|\dot{\eta}(x_0)\|\|\dot{\psi}(x_0)\|.$$

Proof of Lemma 4: By $\eta = \mathbb{P}(Y = 1|X = x) = \frac{\pi_1 f_1}{\pi_1 f_1 + (1-\pi_1)f_0}$, we have

$$\dot{\eta} = \frac{\pi_1(1-\pi_1)(\dot{f}_1 f_0 - f_1 \dot{f}_0)}{(\pi_1 f_1 + (1-\pi_1)f_0)^2}.$$

For $x_0 \in \mathcal{S}$, $\pi_1 f_1(x_0) = (1-\pi_1)f_0(x_0) = \frac{1}{2}\bar{f}(x_0)$, we have

$$
\begin{aligned}
\dot{\eta}(x_0) &= \frac{\pi_1(1-\pi_1)(\dot{f}_1(x_0)f_0(x_0) - f_1(x_0)\dot{f}_0(x_0))}{[\pi_1 f_1(x_0) + (1-\pi_1)f_0(x_0)]^2} \\
&= \frac{1/2(\pi_1\dot{f}_1(x_0) - (1-\pi_1)\dot{f}_0(x_0))}{\bar{f}(x_0)} = \frac{\dot{\psi}(x_0)}{2\bar{f}(x_0)}.
\end{aligned}
$$

Therefore,

$$2\bar{f}(x_0)\|\dot{\eta}(x_0)\| = \|\dot{\psi}(x_0)\| \text{ and}$$
$$\dot{\psi}(x_0)^T\dot{\eta}(x_0) = 2\bar{f}(x_0)\dot{\eta}(x_0)^T\dot{\eta}(x_0) = \|\dot{\eta}(x_0)\|\|\dot{\psi}(x_0)\|.\blacksquare$$

**Lemma 5** *[6] For any distribution function G, constant a, and constant b > 0, we have*

$$\int_{-\infty}^{\infty}\big\{G(-bu-a) - \mathbb{1}\big\{u < 0\big\}\big\}du = -\frac{1}{b}\big\{a + \int_{-\infty}^{\infty} t\,dG(t)\big\},$$

$$\int_{-\infty}^{\infty} u\big\{G(-bu-a) - \mathbb{1}\big\{u < 0\big\}\big\}du$$
$$= \frac{1}{b^2}\big\{\frac{1}{2}a^2 + \frac{1}{2}\int_{-\infty}^{\infty} t^2\,dG(t) + a\int_{-\infty}^{\infty} t\,dG(t)\big\}.\blacksquare$$

**Lemma 6** *[6] Given $\alpha_i = i^{1+2/d} - (i-1)^{1+2/d}$, we have*

$$(1 + \frac{2}{d})(i-1)^{\frac{2}{d}} \le \alpha_i \le (1 + \frac{2}{d})i^{\frac{2}{d}}, \tag{S.36}$$

$$\sum_{j=1}^{k} \alpha_j^2 = \frac{(d+2)^2}{d(d+4)}k^{1+4/d}\big\{1 + O(\frac{1}{k})\big\}.\blacksquare \tag{S.37}$$