[Reviews · NeurIPS 2020]

Review 1

Summary and Contributions: This paper build's on Samworth's optimal-weighted nearest neighbor asymptotic expansion to show that the recently proposed majority-voting distributed nearest neighbor approach (Qiao et al 2019) does not have an optimal multiplicative constant for the variance term in the regret (although its rate is optimal as established previously by Qiao et al). With a simple modification of replacing the majority vote with a weighted voting strategy, this paper shows that the suboptimal multiplicative constant can be removed. This paper also provides a sharp upper bound on the number of subsamples that should be used to obtain the optimal convergence rate.

Strengths: Whereas nearest neighbor theory papers have largely not worried too much about constants, this paper digs into the nuances to get a specific constant from pi/2 down to 1 with a very simple algorithmic idea: use weighted voting rather than simple majority voting when ensembling local nearest neighbor classifier predictions (i.e., for each local estimate, output not just a single label but instead a posterior probability estimate, and then at the end average across these posterior probability estimates and threshold; this is in contrast to outputting a single label per local nearest neighbor classifier and then taking a majority vote across the labels). This analysis is neat albeit limited in scope. I suspect that this work will be of interest mostly to researchers working on theory for nonparametric methods. The experimental results are rather unsurprising though they are reassuring: majority voting vs weighted voting take about the same amount of time, and both offer a speedup compared to regular k-NN; weighted voting has slightly lower risk than majority voting.

Weaknesses: The scope of the analysis is very limited to distributed nearest neighbor classification (along with some distributional assumptions that the authors point out in the discussion section to be a bit restrictive), and it's a bit unclear whether insights here generalize to other methods.

Correctness: The claims and methods appear correct.

Clarity: Currently the paper has lots of small typos. Please proofread carefully and revise. Also, I find Table 1 to be slightly confusing to parse for the risk percentages. Whereas the speedup comparison is clearly defined as a ratio of the slower time divided by the faster time, how is the risk percentage defined in comparison to the oracle KNN/OWNN? Additive?

Relation to Prior Work: The authors have done a fairly thorough literature review.

Reproducibility: Yes

Additional Feedback: I'd suggest adding error bars to Table 1 (for example, to denote standard deviations across experimental repeats). Also, the multi-cohort medical study example could be good to mention in broader impacts (right now the broader impacts instead just reads too much like what your contributions are rather than what the broader impacts are).


Review 2

Summary and Contributions: In this paper, the authors consider the nearest neighbor classification for large-scale data and show that the distributed nearest neighbor classifier achieves the same rate of convergence as its oracle version in terms of regret when majority voting is used up to a multiplicative constant. The authors also show that the multiplicative difference can be eliminated by using the weighted voting scheme instead of majority voting when combining distributed local classifier assignments. Lastly, the authors derive an upper bound on the number of subsamples so that the distributed nearest neighbor classifier reaches to the optimal convergence rate. However, the derivations are done for only two classes (for binary classification) which significantly degrades the importance of these findings and limits its usage in real-world problems.

Strengths: The authors prove some findings that may be important for the researchers working on large-scale distributed nearest neighbor classification. Basically, the following contributions are made: i) It is shown that the distributed nearest neighbor classifier achieves the same rate of convergence as its oracle version in terms of regret when majority voting is used up to a multiplicative constant. ii) The authors also show that the multiplicative difference can be eliminated by using the weighted voting scheme instead of majority voting when combining distributed local classifier assignments. iii) The authors derive an upper bound on the number of subsamples so that the distributed nearest neighbor classifier reaches to the optimal convergence rate. This allows to choose a suitable number of machines that will be used for parallel computing.

Weaknesses: Unfortunately, I strongly believe that this paper will have a very limited attraction from the research community since derivations are done for binary classification and the nearest neighbor classification is no longer popular as before as there are numerous good alternatives. To validate my claim, I looked at the recent Neurips 2019 paper cited as [45] which is quite similar to this paper. In one year, it is cited only once. This is quite natural in my opinion since deep neural networks dominated classification and there are good alternatives to the nearest neighbor classification for large-scale data as hashing, approximate nearest neighbor classification methods, etc. Especially, unsupervised and supervised hashing methods are quite popular for large-scale data with high-dimensional feature spaces. Therefore, I strongly believe that the impact of the paper is very limited and it will attract a very few attention from research community. Also, to demonstrate the advantages of distributed nearest neighbor classification (if there exist any) over competing methods such as hashing and approximate nearest neighbor classifiers (kd tress, FLANN etc), additional experiments are needed. Lastly, advantages of using weighted voting scheme instead of majority voting must be clear since it is already well studied in general classification combination. It is also very intuitive since one can either obtain the same accuracy as majority voting or improve accuracy since we end up majority voting when we use the same weighting for all classifiers. Minor Issue: the authors mention divergence of s, which is the number of subsamples. It is not a sequence, thus I do not understand what the authors meant with this?

Correctness: There are too many mathematical work including propositions, remarks and theorems. I am not a Mathematician, therefore I cannot be sure if they are correct or not since I did not check them very carefully.

Clarity: Paper is written well but there are too many equations and mathematical work which makes very difficult to understand it completely.

Relation to Prior Work: Mostly differences with respect to earlier works are clear. But, it will be nice to write a paragraph summarizing the basic differences with respect to the papers given in [45] and [46].

Reproducibility: No

Additional Feedback: The major limitation of the paper is that all derivations are done for binary classification. The large-scale data generally include many classes and this limits the applicability of the proposed method to real-world problems. Furthermore, nearest neighbor is no longer popular as there are many alternatives including deep neural nets, hashing, etc.


Review 3

Summary and Contributions: The paper studies the regret of nearest neighbor classifier under the distributed setting. In particular, the number of subsamples s is allowed to grow with sample size N. Two types of aggregation scheme are introduced---majority voting (Algorithm 1, M-DiNN) and weighted voting (W-DiNN). The former scheme can achieve the "optimal" convergence rate up to a constant that only depends on the dimension d, while the multiplicative difference can be eliminated by the latter one. Numerical results are provided to illustrate the performance of the proposed method.

Strengths: One major contribution of the paper is that the results tell us under the distributed setting, how data should be split, and whether it is possible to achieve a convergence rate the same as that on the entire data set when s grows with N. The results also tell us how to assign weights on each subsample. Admittedly, the results rely on the recent work on the error rate of the nearest neighbor classifier, such as Tsybakov, 2004 and Samworth, 2012, but I think the authors also make their own substantial contributions.

Weaknesses: The assumptions A1 to A4 are typically assumed when studying the error rate of nearest neighbor classifier. In discussion, the authors also mentioned that some of these assumptions are quite strong, like bounded support and twice differentiability around the decision boundary. It would be interesting to study whether these assumptions can be relaxed. But I think even with these assumptions, the results also give us useful implications, especially for how to split the data.

Correctness: I did not check the details of the proof, but I check the result in certain special cases. They match the existing ones on error rate of kNN.

Clarity: The paper is well written.

Relation to Prior Work: The authors have provided a detailed review of relevant methods.

Reproducibility: Yes

Additional Feedback: My score remains the same after reading the authors response.

[Author Response · NeurIPS 2020]

We thank all 3 reviewers for their thoughtful comments.

**Reviewer 1:** *"nearest neighbor theory papers have largely not worried too much about constants......This analysis is*
*neat albeit limited in scope. I suspect that this work will be of interest mostly to researchers working on theory for*
*nonparametric methods."* In the evolution of the study of nearest neighbor, early work focused on consistency, and later
work goes beyond consistency and focuses on rate of convergence. The logical next step of theoretical interest would be
on the constant. You are absolutely correct that very few work studies the constant. We argue that this is "a feature, not
a bug". The seemingly relative unpopularity of this type of analysis may be due to its technical challenge and depth of
the analysis.

*"The scope of the analysis is very limited to distributed nearest neighbor classification (along with some distributional*
*assumptions that the authors point out in the discussion section to be a bit restrictive), and it's a bit unclear whether*
*insights here generalize to other methods."* Our analysis generalizes to other ensemble methods as well as data-
interpolation weighted methods. The latter is a fairly interesting direction, due to its connection with deep learning. We
leave these explorations as future works.

*"Currently the paper has lots of small typos. Please proofread carefully and revise.."* Thanks for pointing out, and we
will fix the typos in the final version, if accepted.

*"Also, I find Table 1 ... How is the risk percentage defined in comparison to the oracle KNN/OWNN? Additive?"* The
risk in the table are empirical classification error; the risks of our proposed methods and the oracle KNN/OWNN are
calculated separately so that one can compare the numerical values in the table directly.

*"I'd suggest adding error bars to Table 1 (for example, to denote standard deviations across experimental repeats). Also,*
*the multi-cohort medical study example could be good to mention in broader impacts..."* Thank you for the suggestions.
We will add them in the final version, if accepted.

**Reviewer 2:** *"The derivations are done for only two classes (for binary classification) which significantly degrades*
*the importance of these findings and limits its usage in real-world problems."* The proofs to our main results are very
convoluted even for the binary case. In order not to distract from the main message, we choose to state the results in
terms of the binary classification setting. All the results can be naturally generalized to the multi-class setting. We stress
that we are not alone in this choice. Almost all the major theoretical work on nearest neighbor in the last 10 years focus
on binary classification. For example, [46], [11], and [25] cited in the paper.

*"nearest neighbor classification is no longer popular as before as there are numerous good alternatives.... In one year, it*
*([45]) is cited only once. This is quite natural in my opinion since deep neural networks dominated classification...."* Due
to the good interpretability and relative low time complexity, nearest neighbor methods are still of high interest among
the practitioners and the nonparametric community. It is true that deep neural networks dominated the classification
literature but it does not mean that there is no room for other important and widely used methods. In particular, the
sheer volume of works on deep neural networks may be due to the fact that they are relatively new and not very well
understood, especially their statistical guarantees. In contrast, many aspects of nearest neighbor have been studied
and the rest are really difficult to analyze. The latter is the gap we try to fill. Lastly, we stress that deep and insightful
theoretical work in nearest neighbor, such as [46] and [11], are still highly cited (for 234 times and 85 times respectively.)

*"there are good alternatives to the nearest neighbor classification for large-scale data as hashing, approximate nearest*
*neighbor classification methods, etc."* All these acceleration approaches provide approximate (not exact) nearest
neighbor classification. However, there is no statistical guarantees in terms of their learning performance. In practice,
distributed learning can be combined with such approaches to further speed up the learning process.

*"Lastly, advantages of using weighted voting scheme instead of majority voting must be clear since it is already well*
*studied in general classification combination. It is also very intuitive ..."* These may be intuitive, but to the best of our
knowledge, we are the first to rigorously prove them and quantify the multiplicative constant. We also give the exact
loss on accuracy due to the choice of the weighting scheme. This finding is subtle, but important.

*"the authors mention divergence of s, which is the number of subsamples. It is not a sequence, thus I do not understand*
*what the authors meant with this?."* s increases as N (size of the whole dataset) increases. One of our main contributions
includes proving that the sharp upper bounds of s (number of subsamples) are $s \asymp N^{2/(d+4)}$ and $s \asymp N^{4/(d+4)}$ for
M-DiNN and W-DiNN, respectively. In this sense, the s can be considered as a sequence $s(N) \to \infty$ as $N \to \infty$.

*"But, it will be nice to write a paragraph summarizing the basic differences with respect to the papers given in [45] and*
*[46]."* In the introduction section, the 3 paragraphs started from line 52 have summarized our contribution compared to
[45] and [46]. We will re-summarize them in a separate paragraph, in the final version, if accepted.

**Reviewer 3:** We thank you for your very positive comments.

[Meta-Review · NeurIPS 2020]

The paper provides some interesting insights into distributed weighted nearest neighbors methods with some nice theoretical implications. I believe the results would be of interest to the nonparametric statistics community. If accepted, additional experimental results against widely used scalable nearest neighbors approaches would greatly improve the practical impact of the work -- however this suggestion is optional and at the discretion of the authors.